psychology

implicit theory of mind, false belief, replication, ignorance, children

**Author for correspondence:**
Marina Proft
e-mail: mproft@uni-goettingen.de

# Do young children track other's beliefs, or merely their perceptual access? An interactive, anticipatory measure of early theory of mind

Pamela Barone[1,2], Lisa Wenzel[3], Marina Proft[3] and Hannes Rakoczy[3]

[1]Department of Psychology, Universidad Católica de Murcia (UCAM), Campus de los Jerónimos, 30107 Murcia, Spain
[2]Human Evolution and Cognition Group (EvoCog), University of the Balearic Islands, Carretera de Valldemossa km 7.5, 07122 Palma, Illes Balears, Spain
[3]Department of Developmental Psychology, University of Göttingen, Waldweg 26, 37073 Göttingen, Germany

PB, 0000-0002-1273-199X; LW, 0000-0003-1761-0588

This paper aimed to contribute to answering three questions. First, how robust and reliable are early implicit measures of false belief (FB) understanding? Second, do these measures tap FB understanding rather than simpler processes such as tracking the protagonist's perceptual access? Third, do implicit FB tasks tap an earlier, more basic form of theory of mind (ToM) than standard verbal tasks? We conducted a conceptual replication of Garnham & Perner's task (Garnham and Perner 2001 *Br. J. Dev. Psychol.* **19**, 413–432) simultaneously measuring children's anticipatory looking and interactive behaviours toward an agent with a true or FB (N = 81, M = 40 months). Additionally, we implemented an ignorance condition and a standard FB task. We successfully replicated the original findings: children's looking and interactive behaviour differed according to the agent's true or FB. However, children mostly did not differentiate between FB and ignorance conditions in various measures of anticipation and uncertainty, suggesting the use of simpler conceptual strategies than full-blown ToM. Moreover, implicit measures were all related to each other but largely not related to performance in the standard FB task, except for first look in the FB condition. Overall, our findings suggest that these implicit measures are robust but may not tap the same underlying cognitive capacity as explicit FB tasks.

# 1. Introduction

Theory of mind (ToM) is the capacity to understand that oneself and others are agents with mental states such as beliefs, intentions and desires [1]. At the core of ToM lies meta-representation [2]: representing that agents represent the environment from a subjective point of view, in ways that can be divergent both from one's own perspective and from objective facts [3–5]. The litmus test for developing meta-representation is the false belief (FB) task. In the standard change-of-location FB task, an object is hidden in location A while an agent is present, but is then transferred to location B while she is absent [1]. Children are then asked to explicitly predict the agent's behaviour (searching for the object) that is based on her FB. Using many variations of this explicit type of task regarding format, structure or topic, a huge body of converging evidence has shown that the underlying core competence for meta-representation and thus a full-blown ToM emerge by the age of 4 [6].

In the last 15 years, however, new research has challenged this standard picture. Findings with various implicit measures have provided evidence for early and automatic ToM abilities in infants and adults [7–12]. Most of the measures are non-verbal and include violation-of-expectation (VoE) looking time [7,13,14], anticipatory looking [8,12,15] and spontaneous interaction [11,12,16,17]. Yet, some of the novel tasks also did involve some linguistic elements and demands. For instance, in a verbal preferential-looking task, children listened to a FB story while looking at a picture book that contained matching and non-matching pictures at the end of the story [18]. Children looked longer at the picture depicting the agent searching the object in the original than at the current location, suggesting FB tracking. In another study, an agent interacted with a child and then left a pair of scissors inside a box. After she left the room, the experimenter changed the scissors from the box to his pocket and asked aloud, looking at the ceiling, 'Where will she think they are?' [19]. Following this prompt, children in the FB condition looked reliably longer at the scissors' original location (the box) than children who received the same question but directed to them (i.e. the experimenter looked directly at the child).

If reliable and valid, these data from the new wave of implicit tasks would have far-reaching theoretical implications: Nativist accounts take such findings to show an early, full-fledged, meta-representational competence to attribute beliefs. It is just that this competence is masked early in development, by (linguistic, executive, etc.) performance factors in the standard FB tasks [20–24]. Two-systems accounts take these findings to tap an implicit, automatic, efficient and early developing mind-reading system that precedes the ontogenetic emergence of the fully fledged, meta-representational and flexible ToM employed in the classical, explicit FB tasks (e.g. [25]).

However, a recent replication crisis raises the question whether findings on early and automatic FB understanding really are reliable and valid. Regarding their reliability, a growing number of non-replications complicates the interpretation of the original findings [26]. A recent meta-analysis of infant FB tasks showed that most positive evidence was found in VoE paradigms. However, conceptual replications of VoE studies have yielded mixed, but mostly negative results [27–29]. Similarly, both direct and conceptual replications of anticipatory looking paradigms also mainly failed to produce the original findings in infants, children and adults [27,30–33] (but see [34,35]) and even interactive FB tasks failed to provide robust and replicable results showing mixed findings [27,31,36–38] (see also [39,40] for a debate regarding the interpretation of failed replications of infants FB studies).

Regarding validity, most replication studies have yielded the following picture [28,32]: first, those implicit tasks that clearly tap meta-representational ToM capacities tend to be not reliably replicable (e.g. [28,38]). Second, those tasks that are replicable, are subject to more parsimonious explanations (e.g. [41]). And third, implicit tasks that clearly require meta-representations could not be solved earlier than explicit tasks (e.g. [31,42]). Furthermore, only a few studies have tested for convergent validation among implicit measures, and the few existing studies mostly failed to find any evidence for converging validity [27,29,43,44]. In stark contrast, superficially different explicit ToM tasks have been shown to correlate and converge, indicating that they all tap the same underlying phenomenon [45,46].

Against this background, the present study focuses on three central questions. First, how robust and reliable are different implicit measures of early ToM? To this end, we implemented a conceptual replication of a study that combined looking and interactive measures [34]. In that study, 2.5- to 4-year-old children were tested in a standard and an interactive FB task. In the scenario, a protagonist lived in a puppet house with two slides, each of them leading to one box. An object was put in box 1 and then transferred to box 2 in the presence (true belief, TB) or absence (FB) of the protagonist. In the standard version of the task, children were asked to anticipate which slide the protagonist would come down to look for her object. In the interactive version, they had to anticipate the protagonist's

behaviour by moving a mat in order to help her land safely at the end of one slide. In addition, children's looking behaviour was measured in every version of the task.

The authors scored the pattern of children's answers for each response (preferential look, mat-moving and verbal behaviour) in FB and TB scenarios. They scored A when the responses were directed to the empty box, B to the box with the object or N if there was no preference for one box over the other. Consequently, AB pattern (responding A in the FB scenario, and B in the TB condition) would be evidence of belief understanding. Results revealed that children showed the AB pattern more often than the BA pattern (considered as random responding) in their interactive and anticipatory-looking responses and, thus, correctly anticipated the agent's behaviour. However, they were not able (yet) to show this response pattern through their explicit answers—suggesting an implicit understanding of belief-based actions in looks and deeds that did not (yet) manifest itself in words [34]. Nevertheless, although the frequency of the AB pattern was higher than the BA pattern in children's looking and interactive responses, half of the participants (24/47) indeed showed a BB response in the mat-moving condition, i.e. they moved the mat to the full box both in FB and TB scenarios.

The second question of interest is the following: do these measures actually tap FB understanding or merely a simpler process such as tracking the protagonist's perceptual access [47,48]? In particular, may children have acted on the basis of a simple 'seeing = knowing rule' (if an agent has seen an event she is knowledgeable and thus able to act successfully; in contrast, if she has not seen the event, she is ignorant and thus not able to act successfully [48]) rather than by belief ascription? To address these questions, we implemented a new ignorance (IG) control condition, in which the protagonist neither witnesses the placement of the object in one box nor its transfer to the other box. The contrast between FB and IG conditions in terms of anticipatory looking (first look and differential looking score), interactive behaviour (children's catch behaviour at the end of one slide) and signs of uncertainty (latency to perform the catch behaviour and number of changes in making the decision) helps to decide between the two alternatives: if children merely operate with perception tracking and a simple 'seeing = knowing rule', they should differentiate FB from TB, and IG from TB, but not FB from IG for all three measures (since the protagonist does not know relevant facts in both conditions, children should simply fail to show any systematic anticipation to one side or the other and thus feel equally uncertain and hesitant to choose one side). By contrast, if children do ascribe proper beliefs, they should perform in markedly different ways in TB (showing anticipation toward the full box), FB (showing anticipation toward the empty box) and IG (showing no such preferential anticipation) for the interactive behaviour measures. For the uncertainty measures we would not expect a difference between TB and FB (since the child should be certain in her response that the agent will come down the slide of the full or empty box, respectively), but a strong effect for the TB/IG and FB/IG comparisons (since the child should hesitate in her response in the IG condition in which there is no determinate expectation and no correct response, so the child can thus only guess).

Third, do infant (interactive and anticipatory looking) FB tasks really tap an earlier and more basic form of ToM than standard verbal tasks? Or do the two kinds of tasks converge such that children solve both only from around age 4 (as suggested in some recent studies, [31,38])? To address this question, children's performance in the target (anticipatory looking and interactive) task was analysed as a function of their performance in a standard verbal FB task.

In order to contribute to answering these three questions, we tested two age groups of children at the ages of 3 and 4, supposedly before and after the critical 4-year transition.

In summary, we devised a conceptual replication and theoretically motivated extension of the original study of Garnham & Perner [34] because this type of task allowed us to do several crucial things at once: to combine different implicit measures (anticipatory looking and interactive), to add a crucial ignorance condition, and to compare children's performance with a standard FB task—all of which together make this task particularly informative.

## 2. Material and methods

### 2.1. Participants

The final sample consisted of 81 children (35 girls, 50 children around their third birthday (age range = 33 to 39 months; mean age = 35.6 months) and 31 children around their fourth birthday (age range = 45–51 months; mean age = 47.78 months); see appendix A for sample size calculation). Twenty-eight additional children were tested but had to be excluded from the analyses (for details, see appendix A). Participants

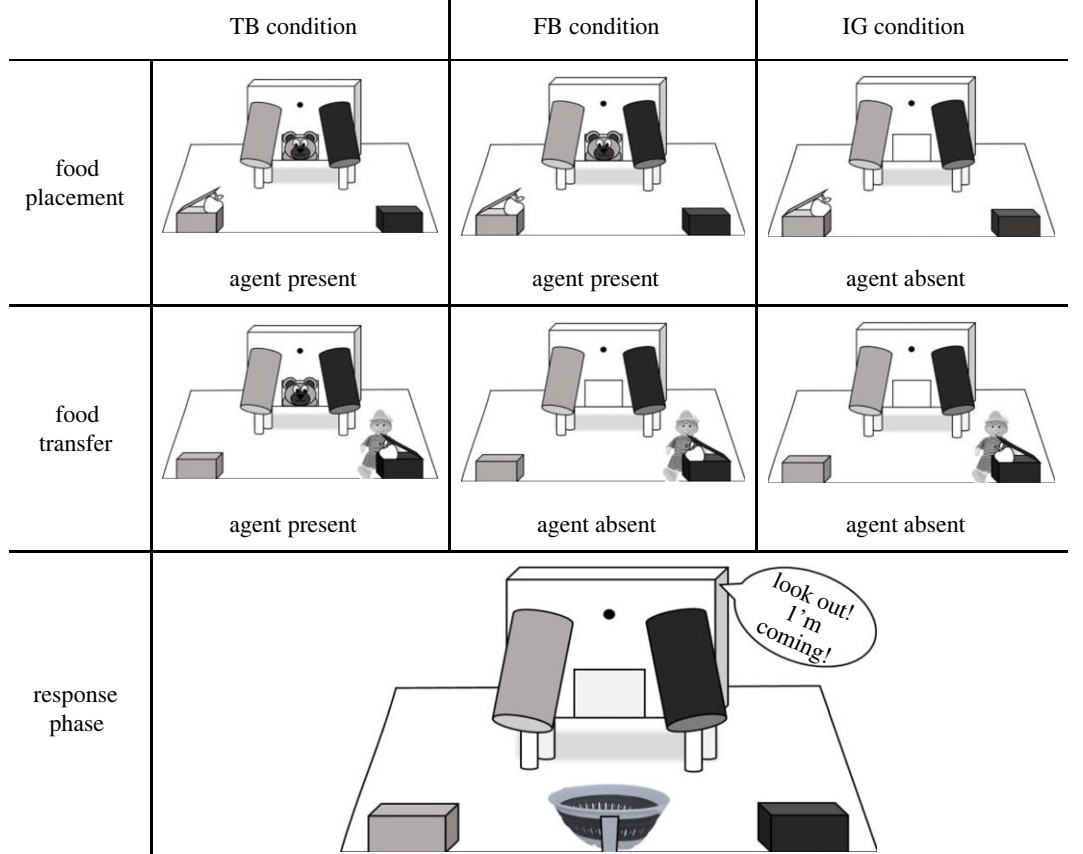

|  | TB condition | FB condition | IG condition |
|---|---|---|---|
| food placement | agent present | agent present | agent absent |
| food transfer | agent present | agent absent | agent absent |
| response phase | | | |

look out! I'm coming!

**Figure 1.** Test conditions in the interactive change-of-location task. TB, FB and IG conditions varied according to the bear's perceptual access (presence or absence) to the food placement and transfer.

were recruited from a database of children whose parents had previously given consent. Children were tested by two female experimenters in the lab.

## 2.2. Design and procedure

All children received two tasks: first, an interactive change-of-location task with condition (TB, FB, IG) as within-subjects factor (one trial per condition; order counterbalanced) and a standard FB task at the end.

### 2.2.1. Interactive change-of-location task

The interactive change-of-location task consisted of a demonstration phase (two trials), a familiarization phase (four trials) and a test phase with one test trial per condition (TB, FB and IG; order counterbalanced), interspersed with filler trials after each condition.

The child sat in front of a table that had a puppet theatre on it: a house with a yellow and a pink slide, and two boxes at the end of each slide (figure 1 for a schematic depiction and appendix A for a full description). Experimenter 1 (E1) sat next to the setting (side of E1 was counterbalanced). The task started with E1 introducing the main protagonist, i.e. a bear (played by Experimenter 2; E2) and explaining the basic procedure: if food is put in one of the coloured boxes, the bear will come down the corresponding same-coloured slide (yellow box/yellow slide, pink box/pink slide).

#### 2.2.1.1. Demonstration phase

In the first demonstration trial, the bear looked through the curtain in the bottom centre of the house and stated that she was very hungry. E1 offered, together with the child, a wooden food item and placed it in one of the boxes, e.g. the pink one. The bear stated that she was going to get the food and therefore was going to come down the appropriate slide (e.g. the pink slide). She went inside the house, walked up the stairs (accompanied by walking sounds) and stated, 'Look out, I am coming!'. After 3 s, the bear came

down the corresponding slide. To increase children's active engagement in future trials, the negative consequences of failing to intervene appropriately were demonstrated: the bear fell onto the table, hurt herself and did not gain the food item. After comforting her, the bear trudged back into her house. A second demonstration trial was conducted for the other slide.

### 2.2.1.2. Familiarization phase

The familiarization phase consisted of four trials (two trials per slide in partially counterbalanced order[1]) in which the child's active intervention of catching the bear was introduced. In the first familiarization trial, E1 hid the piece of food in one box, then brought a sieve into the scenario and showed how to use it under the slides to catch the bear when she slid. Both the child and the bear agreed on cooperation, so the bear went inside the house in order to take the corresponding slide. While the bear's walking sounds were audible, E1 handed over the sieve to the child. The bear stated, 'Look out, I am coming!' and, after 3 s, she came down the corresponding slide.

If the child put the sieve under the correct slide, the bear was caught and fed; if the sieve was held under the incorrect slide, the bear fell onto the table, got hurt and trudged back home without eating (like in the demonstration phase).

### 2.2.1.3. Test trials

Each child performed one test trial in each condition (TB, FB and IG; order counterbalanced, figure 1). Each trial started with the bear at the door, claiming she was hungry. In the FB condition, for example, the bear watched E1 placing a food item in one of the boxes, then stated that she was going to get it and disappeared inside the house. While she was absent, a second hand puppet (Tom) appeared, transferred the food item to the other box and left the scene. E1 made a remark about the bear's perceptual access to the transfer ('The bear hasn't seen what Tom did, has she?') and reacted according to the child's answer (confirmative, if the child answered correctly; corrective if the child answered incorrectly).

Conditions varied according to which critical information about the location of the food was available to the bear. In the TB condition, the bear witnessed both placement and transfer of the food. In the FB condition, she saw the placement of the object but did not see the transfer. In the IG condition, the bear did not witness placement or transfer, thus she was ignorant about the food's location. To make it clear to the child that in the IG condition the bear did not witness the original placement and transfer, E1 brought the item of food to the main stage, but before placing it in one of the boxes, the bear said 'Bye' and disappeared from the scene. In the bear's absence, E1 stated 'I put it in here' and placed the food in one box. Still in the bear's absence, Tom appeared and transferred the food. After the transfer, E1 said the same prompt as in the FB condition ('The bear hasn't seen what Tom did, has she?') and reacted according to the child's answer.

In all conditions, when the bear made walking sounds, E1 handed over the sieve and the bear stated, 'Look out, I am coming!' (the anticipation signal). In order to prevent learning effects in the test trials, the actual outcome of the bear sliding was never shown in the test trials. Instead, E1 waited for 3 s in which the child could make a decision and then shouted toward the bear that she found another, more desirable food item for her (e.g. chocolate). Hearing this, the bear did not slide and went down the stairs again to have a look through the door.

### 2.2.1.4. Filler trials

Each test trial was followed by a filler trial, taking the same procedure as familiarization trials: the bear saw in which box the most valuable piece of food (e.g. chocolate) was hidden, went inside the house, and made walking sounds. E1 handed over the sieve to the child, the bear stated, 'Look out, I am coming!' and, after 3 s, she came down the corresponding slide.

### 2.2.2. Standard FB task

The standard FB task (after [1]) was conducted by E2. First, E2 introduced the agent (a stuffed dog) who placed her ball in one of two empty boxes. In her absence, a second agent (a monkey) transferred the

---

[1]Children received one of these four possibilities: (i) pink-yellow-pink-yellow, (ii) pink-yellow-yellow-pink, (iii) yellow-pink-yellow-pink, or (iv) yellow-pink-pink-yellow.

object to the other box. After the transfer, E2 asked an explicit test question ('When the dog comes back, where is he going to look for his ball first?'), followed by three control questions (CQ1: 'Where did the dog put his ball in the beginning?', CQ2: 'Where is the ball now?', CQ3: 'And did the dog see what the monkey did?').

## 2.3. Coding

In the interactive change-of-location task, we coded the following dependent variables:

Anticipatory looking: we coded children's first look toward the slides/boxes after the bear said 'I'm coming'. In addition, following many anticipatory looking studies (e.g. [49]), a differential looking score (DLS) was calculated by subtracting the total looking time to the empty side (slide and box) from the total looking time to the full side (slide and box), and dividing it by the sum of looking time to both sides (see appendix A for formula). The resulting DLS measured children's preference to look at one side over the other (maximum of preference to the full side = 1, maximum of preference to the empty side = −1, no preference = 0).[2]

Anticipatory behaviour (i.e. catch): whether children put the sieve under the slide that led to the box with food (hereinafter referred to as 'full box') or under the slide that led to the box that previously contained the food (i.e. 'empty box').

Measures of uncertainty: we coded the latency of the anticipatory behaviour, that is, the time children took to start the catch behaviour after the sieve was available. We also calculated catching and gaze shifts, that is, the amount of changes in children's catching and looking behaviour (going back and forth from one slide to the other).

In familiarization and filler trials, we coded children's anticipatory behaviour. In the standard FB task, we coded which box (full or empty) the child referred to after the test question.

# 3. Results

The main results from the test trials are depicted in figure 2 (for results from familiarization and filler trials, figures with the results as a function of age group, first trial analyses, results as a function of standard FB task performance, and for additional details, see appendix B).

## 3.1. Anticipatory looking

### 3.1.1. First look

A binomial logistic regression was computed to analyse the effect of condition on the likelihood that participants looked first at the full box. FB condition was set as the reference category because we were interested in the FB–TB and FB–IG comparisons. The logistic regression model was statistically significant, $\chi^2_{239} = 20.798$, $p < 0.001$. Participants were more likely to look first at the full box in the TB condition than in the FB condition ($\beta = 1.45$, $Z_{239} = 4.219$, $p < 0.001$, OR = 4.28). However, children in the IG and FB conditions were as likely to look first at the full box ($\beta = 0.323$, $Z_{239} = 1.021$; $p = 0.307$, OR = 1.38).

Analysed separately, children directed their first look significantly more often than chance to the full box (77%; binomial test, $p < 0.001$) in the TB condition. In FB and IG conditions, however, children performed at chance level (43% and 51% looked at the full box first, respectively; binomial tests, $p$s > 0.05).

### 3.1.2. Differential-looking score

Given the numerical nature of DLS data (ranging from −1 to 1), we computed a one-way repeated-measures ANOVA to assess whether there was an effect of condition in children's DLS. Mauchly's test of sphericity indicated that the assumption of sphericity had not been violated, $\chi^2_2 = 0.988$, $p = 0.61$. The analysis showed that DLS differed significantly between conditions, $F_{2,160} = 5.306$, $p = 0.006$, $\omega^2 = 0.032$ suggesting that this is a small to medium effect.

---

[2]Garnham and Perner [34] coded anticipatory looking in a slightly different way. They considered a 3 s period and coded which side (location A, location B or no preference) each child looked longer in each condition. Instead, we consider looking behaviour both as a categorical (which side children looked first) and numerical variable (by calculating the DLS).

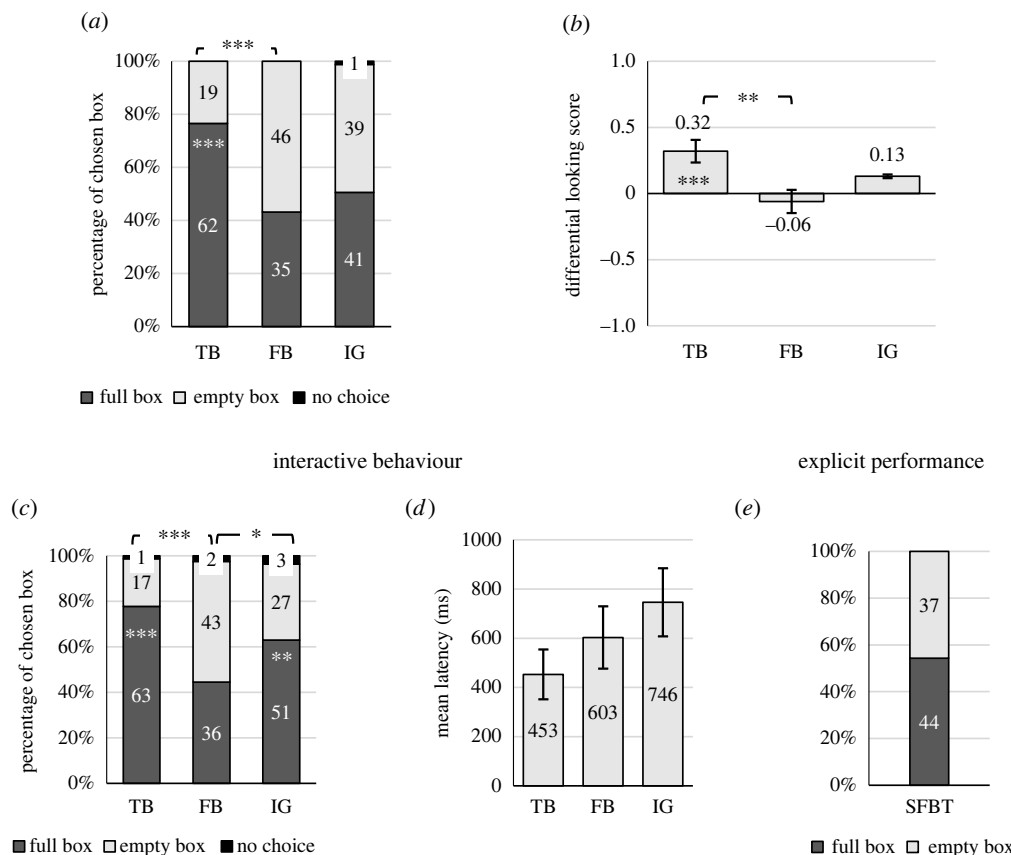

**Figure 2.** Results of the interactive FB task: (a) proportion of children's first look at each box (full/empty box), (b) differential looking score, (c) proportion of children catching the agent under each box (full/empty), (d) latency of children's interactive behaviour and (e) proportion of children choosing each box (full/box) in the standard FB task. $^{*}p < 0.05$; $^{**}p < 0.01$; $^{***}p < 0.001$.

*Post hoc* testing using the Bonferroni correction revealed that DLS was significantly higher in TB ($M = 0.325$, s.d. $= 0.77$) compared with FB condition ($M = −0.06$, s.d. $= 0.79$), $p = 0.004$, $d = 0.49$. However, DLS was not significantly different between TB and IG conditions ($M = 0.13$, s.d. $= 0.778$), $p = 0.274$, $d = 0.26$; or between FB and IG conditions, $p = 0.363$, $d = 0.24$.

Separate one-sample *t*-tests indicated that participants preferred to look at the full box in the TB condition, $t_{80} = 3.796$, $p < 0.001$, $d = 0.42$; but showed no preference to look at one box or the other in the FB condition, $t_{80} = −0.066$, $p = 0.51$, $d = −0.07$, or in the IG condition, $t_{80} = 1.45$, $p = 0.15$, $d = 0.16$.

## 3.2. Interactive behaviour

A binomial logistic regression was computed to analyse the effect of condition on the likelihood that participants' interactive behaviour was directed to the full box. The logistic regression model was statistically significant, $\chi^2_{234} = 19.321$, $p < 0.001$. Participants were more likely to choose the full box in the TB condition ($\beta = 1.488$, $Z_{239} = 4.195$, $p < 0.001$, OR = 4.43) and in the IG condition ($\beta = 0.814$, $Z_{239} = 2.48$, $p = 0.013$, OR = 2.26) than in the FB condition.

Analysing each condition separately, children chose the full box significantly above the chance level in TB and IG conditions (79% and 65% respectively; binomial tests, $p$s < 0.009). In the FB condition, children performed at chance level (46% chose the full box; binomial test, $p = 0.50$).

## 3.3. Measures of uncertainty

Kolmogorov–Smirnov tests indicated that the variables measuring uncertainty (latency and shifts in interactive and looking behaviour) do not follow a normal distribution (all $p$s < 0.001). Therefore, non-parametric tests (Friedman's ANOVA) were performed on such data.

### 3.3.1. Latency

Friedman's ANOVA showed that children's latency to make a decision did not differ between conditions, $\chi_2^2 = 5.27$, $p = 0.072$ (TB: $M = 453.17$ ms, s.e. = 101.29; FB: $M = 603.37$ ms, s.e. = 126.69; IG: $M = 745.93$ ms, s.e. = 138.43).

### 3.3.2. Shifts in interactive behaviour

Friedman's ANOVA showed that the number of shifts in the interactive behaviour did not differ between conditions, $\chi_2^2 = 1.46$, $p = 0.48$. Children switched their catch behaviour similarly between the two slides in the different conditions (TB: range of shifts: 0–1, $M = 0.16$, s.e. = 0.04; FB: range of shifts: 0–3, $M = 0.19$; s.e. = 0.05, IG: range of shifts: 0–2, $M = 0.26$, s.e. = 0.06).

### 3.3.3. Shifts in looking behaviour

Friedman's ANOVA showed that the number of gaze shifts did not differ between conditions, $\chi_2^2 = 3.54$, $p = 0.17$. Children switched their gaze similarly between the two slides in the different conditions (TB: range of shifts: 0–4, $M = 0.65$, s.e. = 0.1; FB: range of shifts: 0–5, $M = 0.93$, s.e. = 0.128; IG: range of shifts: 0–4, $M = 1$, s.e. = 0.126).

## 3.4. Standard FB task

Overall, children performed at chance level in the standard FB task (46% correctly referred to the empty box; binomial test, $p = 0.505$). A chi-square test of independence showed that the relationship between children's performance on the explicit FB task and age group was significant, $\chi^2(1, N = 81) = 24.745$, $p < 0.001$.

A logistic regression was also computed to analyse the effect of age group (younger versus older) on the likelihood that participants chose the full box. The logistic regression model was statistically significant, $\chi_{79}^2 = 26.114$, $p < 0.001$. Participants in the older group were more likely to choose the empty box than participants in the younger group ($\beta = -2.580$, $Z_{79} = -4.587$, $p < 0.001$, OR = 0.076).

Considered separately, younger children failed the standard FB task (76% incorrectly chose the full box; binomial test, $p < 0.001$) and older children passed the test by selecting the empty box significantly more often than chance (81%; binomial test, $p < 0.001$).

## 3.5. Correlations among measures

First, correlations between the different measures of the interactive task (first look, DLS and catch behaviour) were calculated. Within each condition (TB, FB and IG), children's first look, DLS, and interactive behaviour were positively correlated (all correlations were statistically significant, $p < 0.01$; table 1).

Second, we also calculated the correlations between the implicit measures (first look, DLS and catch behaviour) and children's performance in the standard FB task. Only children's first look in the FB condition was associated with their performance in the standard FB task (Cramer's $V = 0.25$, $p = 0.025$), indicating that those children who first looked at the empty slide in the interactive task also passed the explicit FB task. All other correlations were not significant. (For more details, see appendix B).

# 4. Discussion

The present study aimed to shed some light on three main questions in current ToM research. First, how robust and reliable are anticipatory looking and interactive measures of early FB understanding? Second, do these measures actually tap FB understanding? Third, do they really tap an earlier and more basic form of FB understanding than explicit standard tasks? The main findings of our task, in relation to these three questions, were the following: first, we successfully replicated the findings of Garnham & Perner [34]. Children's looking, both first anticipatory look and relative anticipatory looking time (indicated in the DLS), and catch behaviour differed between TB and FB conditions. While performance was at chance level in the FB condition, children correctly looked and actively chose the full box in the TB condition predominantly. Second, children behaved similarly in FB and IG conditions (with the exception of interactive behaviour, in which a significant difference between these two conditions was found). Third, the various measures in the interactive task (first look, DLS, interactive behaviour) were all related to each other, but not to performance in the explicit FB task (except for first look in the FB condition

**Table 1.** Correlations between implicit measures.

| | | DLS[a] | catch[b] |
|---|---|---|---|
| first look | TB | 0.701** | 0.818** |
| | FB | 0.796** | 0.591** |
| | IG | 0.779** | 0.546** |
| catch | TB | 0.798** | |
| | FB | 0.758** | |
| | IG | 0.718** | |

[a]Point-biserial correlations.
[b]Cramer's *V*.
**$p < 0.01$.

which correlated with the explicit FB task). At the same time, these patterns in the various measures in the interactive task (such that children performed differently in TB as compared with FB/IG) could already be found in younger children who still failed standard FB tasks (see appendix B).

Overall, our findings thus show that the results of Garnham & Perner [34] are robust and reliable: for different types of measures (anticipatory looking and interactive behaviour), young children who do not yet master standard FB tasks responded differentially according to the agent's true or false belief. However, the additional results from a new IG control condition suggest that the replicated pattern of findings (in TB versus FB conditions) does not necessarily reveal full-fledged belief understanding. In this IG control condition, the agent failed to witness not only the object's transfer (as in the FB condition) but also its placement in the first place. Thus, the agent was not mistaken about its location, but simply completely ignorant of the object's whereabouts. If children use proper FB ascription in belief tasks, their performance in FB and IG conditions should differ as they should form a clear prediction (indicated by anticipatory looking and catch behaviour) in the FB condition but not in the IG case. Additionally, children should be more uncertain in the IG condition by showing more latency or shifts in their gaze and catching behaviour. If children, however, use a more parsimonious ignorance ascription strategy in FB situations (a 'seeing = knowing' rule of the form: if an agent has seen an event, she is knowledgeable and will act successfully; otherwise, no such expectations hold [50]), the following pattern is presupposed: performance between FB and IG conditions should not differ and similar signs of uncertainty should be present in both conditions. In line with the latter possibility, our results show that only one of these three effects is significant. Indeed, catch performance differed between FB and IG conditions, indicating that children had a preference in the IG condition for choosing the full box. Results presented in appendix B suggest that this behaviour might be dependent on children's performance on the standard FB task: non-passers showed this pattern, while the group of passers chose the two boxes at random in the IG condition.

Still, it might be argued that children's at-chance performance in the FB condition cannot only be interpreted as an indicator of ignorance ascription, but it may rather obscure an underlying pattern of systematic failure and success in the FB task. While half of the children may possess fully developed ToM abilities and therefore answered correctly, the other half answered incorrectly due to their lack of meta-representational ToM capacities. If this were the case, one would expect systematic correlations of the interactive and the standard FB task (such that half of the children fail both, while the other half master both). But there was no such inter-task correlation (once age was controlled for). Another alternative interpretation of children's at-chance behaviour at the group level in the interactive ToM tasks may be that children were somehow overwhelmed by the tasks and generally just guessed. However, this seems not very plausible in light of the fact that children did perform systematically and correctly in the TB version of the task.

Another potential concern regarding the present study is that the IG condition may have been confusing. On the one hand, the bear does not witness any of the object locations and relocations and is thus introduced as naive with respect to the object's whereabouts. However, simply preventing the bear from witnessing the hiding process would suffice for an IG condition. On the other hand, the way E1 and the bear interact may be taken to suggest that E1, something like an omniscient narrator in the scene, expects the bear to know where the object is. In fact, the bear could hear from behind

the scenes when, at the end of each test trial, E1 announced that a more desirable food item was available to her. Hearing this, the bear did not slide. In line with this, it could be argued that the bear possibly heard (and thus knew) that Tom appeared and did something in her absence as well as the prompts that E1 directed to the child. All this may have been a confusing combination for the child. In addition, children in our IG condition were put in a difficult position: we forced them to make a choice even though there was no correct answer. In the end, they chose the full box more frequently than the empty box. One possibility is that this pattern resulted from some form of 'pull of the real': even though the agent does not know about it, there is an objectively good reason to choose the full box as it actually contains the reward (see e.g. [45]). And in cases of uncertainty, it might be safer to go for the objectively good than for the objectively bad answer. Future studies should thus aim at implementing clearer IG conditions by just preventing the agent from witnessing the hiding process and removing the transfer process, by keeping verbal interaction to the minimum or by not forcing the child to give an answer when there is no correct one.

Furthermore, when reporting null results (as in our case no difference between FB and IG in most of our dependent variables) there is always the question of whether the sample size was big enough to detect a potential small effect. As reported in appendix A, our final sample size was based on a small effect (the smallest effect of the original study [34]). It is to note, however, that the original study did not include an IG condition and did not explicitly compare FB and TB conditions. That being said, as we did not find differences between the FB and IG conditions for five of our six measures (not even for the uncertainty measures where on the basis of belief attribution we would expect a big difference between FB and IG), we believe that these results are not masked by a power issue. In spite of that, absence of evidence is not evidence for absence, and to fully exclude this issue an approach of Bayes null hypothesis testing with a larger sample would be the more stringent method for future research.

All in all, and keeping these caveats in mind, the present findings are most compatible with a more parsimonious explanation of children's looking and interactive behaviour as based on tracking information access versus ignorance (rather than full-fledged meta-representational belief ascription). This is very much in line with other recent findings from replication and validation studies of implicit ToM tasks. Several studies found that those tasks that clearly require meta-representational ToM could not be replicated, while those tasks that could be replicated were subject to more parsimonious explanations, for example in terms of knowledge/ignorance tracking (e.g. [28,38,41,51]).

All things considered, the results from the present study draw a complex picture of children's early non-verbal ToM capacities. Converging evidence from anticipatory looking and spontaneous interaction behaviour (in TB conditions, on the one hand, versus FB and IG conditions, on the other) shows that children reliably engage in a form of mental state ascription: they expect an agent to act on the basis of some kind of epistemic state. The results from a new IG control condition, however, in which children largely behave like in the FB condition, suggest that the epistemic state ascription does not amount to full-fledged belief attribution. Rather, children probably merely track knowledge versus ignorance. Furthermore, these basic and implicit ToM capacities seem not to be the same ones as those tapped in standard explicit FB tasks, indicated by the lack of correlation between both kinds of tasks. Additionally, the overall response pattern in the interactive task did not systematically differ if we consider children's performance on the first trial of the task, or between standard FB task passers and non-passers (see appendix B for the results and more thorough discussion).

This general picture is broadly compatible with several recent conceptual change and dual-process accounts (e.g. [25,45,50,52]). Though differing substantially in focus and content, the common denominator of these approaches is their shared claim that in early, basic and precocious ToM, the ascription of epistemic states is primarily factive and relational: with this conceptual framework, children can keep track of who had access to which kind of information, but they cannot yet represent mis-representation. Whether early ToM is indeed best described by some such account, and if so, by which, will be a central question for future research in this area. Another central question for future research concerns the contrast between implicit and explicit tasks. While there are intuitively clear and prototypical tokens of each type (mere looking measures on the implicit, and verbal judgement tasks on the explicit side), (inter)action measures occupy a kind of middle ground. For such measures of spontaneous (inter)action it is much less clear how they align conceptually and empirically (see, e.g. [53–55]). Conceptually, some interaction tasks—such as those used in comparative research [56,57]— clearly require conceptualized thought and rational action planning and thus count as explicit according to most approaches; and empirically, these tasks do converge with standard verbal explicit tasks in children's performance [56,58]. In the current paper, (inter)action measures correlated with looking measures but not with children's verbal responses on the standard task. Future research will

thus need to aim at delineating more clearly, conceptually and empirically, which types of (inter)action tasks align with prototypical implicit ones, and which with explicit ones.

Ethics. This research was conducted in accordance with the 1964 Helsinki Declaration and its later amendments or comparable ethical standards. It involved no invasive or otherwise ethically problematic techniques and no deception (and therefore, according to National jurisdiction, did not require a separate vote by a local Institutional Review Board; see the regulations on freedom of research in the German Constitution (§5(3)), and the German University Law (§22)). Before the study started, informed consent was obtained from the parents of the participants.

Data accessibility. The data generated in this study are available at https://osf.io/j7m2x/.

Authors' contributions. P.B.: conceptualization, data curation, formal analysis, investigation, methodology, writing—original draft, writing—review and editing; L.W.: conceptualization, data curation, formal analysis, investigation, methodology, writing—original draft, writing—review and editing; M.P.: conceptualization, methodology, supervision, writing—original draft, writing—review and editing; H.R.: conceptualization, investigation, methodology, supervision, writing—original draft, writing—review and editing.

All authors gave final approval for publication and agreed to be held accountable for the work performed therein.

Conflict of interest declaration. We have no competing interests.

Funding. The work of Pamela Barone was supported by a PhD scholarship and an inter-institutional research exchange grant from the Spanish Ministry of Economy and Competitiveness (grant no. BES-2014-067640) and from the Spanish Ministry of Science, Innovation, and Universities (grant no. EEBB-I-18-12981), respectively. This work was funded by the Deutsche Forschungsgemeinschaft (DFG, German Research Foundation) – RA 2155/7-1.

Acknowledgements. We would like to thank Marlen Kaufmann and Konstanze Schirmer for their help in recruiting children, as well as Isabel Ganter, Lina Graumann, Joana Lonquich and Leslie-Ann Eickhoff for their help with data collection and coding. We are grateful for the participation of all children and their parents. We further would like to thank Antoni Gomila, the members of the Department of Developmental Psychology at the University of Göttingen, and attendants to the European Conference for Cognitive Science (EuroCogSci 2019) and the 10th annual Budapest CEU Conference on Cognitive Development (BCCCD 2020) for their valuable comments on previous drafts of this manuscript.

# Appendix A. Participants and set-up

## A.1. Participants

We conducted statistical power analyses for sample size estimation (using G*Power 3.1.9.2; [59], aiming at alpha = 0.05 and power = 0.95, based on data from the original interactive FB task study ($N = 47$; [34]). The effect size in the original study for the above-chance performance in the interactive FB task was 0.5. The projected sample size needed with this effect size is approximately $N = 52$.

The smallest effect size in the original study for children's above-chance performance in looking behaviour was OR = 4.5. The projected sample size needed with this effect size is approximately $N = 67$. For the comparison of first look and explicit FB task performance, the effect size in the original study was OR = 3.5. The projected sample size needed with this effect size is approximately $N = 44$. For comparison between interactive behaviour and explicit answer, however, the effect size was OR = 4.3 in the original study, resulting in a needed sample size of $N = 77$. Then, we ended with a final sample size of 81 children.

The IG condition and the hesitance measures were not included in the power calculations since they were not performed in the original study [1] which served as the basis for the power analysis.

Additional 27 children were tested but were excluded from the final analyses because they passed less than three familiarization trials ($N = 8$), refused to cooperate ($N = 10$), did not anticipate the agent's behaviour ($N = 2$), did not understand how to use the sieve ($N = 1$), due to experimental error ($N = 3$), interference by parents ($N = 2$) and technical issues ($N = 1$).

## A.2. Procedure

### A.2.1. Interactive change-of-location task

After entering the testing room, the child sat on a chair in front of the table that held the puppet theatre and E1 sat on a chair next to the child. If the parent also came inside the room, his/her chair was behind the child avoiding direct facial contact.

First, E1 introduced the scene to the child, 'Look [child name], in this house lives a bear which is always very, very hungry. And we are going to feed the bear. Let's call the bear. Bear!! Bear!!' After calling her, the bear appeared at the door at the bottom of the house and greeted the child and E1. Then, E1 introduced the procedure to the child and the bear, 'So, when the bear is hungry, we will

put food in the yellow box and she will take the yellow slide to get it. But if we put food in the pink box, she will take the pink slide'. Immediately, E1 asked the child, 'So, if we put the food in the pink box, which slide do you think the bear will take?' After the child answered, she repeated the question with the other slide, 'If we put the food in the yellow box, which slide do you think the bear will take?'. E1 reacted confirming if the child's answer was correct or correcting if the answer was incorrect. The order of the questions was counterbalanced.

Then, the demonstration phase began. In the first demonstration trial, the bear claimed, 'I am hungry, I am hungry!' So, E1 offered the bear an item of food, 'Ok bear! Look, what we have here for you! It's a [name of the object]. Do you want some bear?' The bear replied, 'Mmm, I would like to have a [name of the object]!' E1 placed the object in the yellow box (order counterbalanced) while saying, 'Okay. I will put the [name of the object] here in this yellow box for you. So, in order to get the [name of the object] you have to go down the yellow slide, okay?' E1 closed the box's lid. The bear replied, 'Yeah, sure, I know! I am going to get the [name of the object] now. Bye!'. The bear went inside the house and walked up the stairs making walking sounds. E1 pointed to the child, 'The bear is going up the stairs!' Then, the bear stated, 'Look out, I am coming! I am coming!'. After 3 s, the bear came down the corresponding slide saying 'Wiii…!'. The bear fell onto the table and hurt herself, 'Ouch, ouch'. E1 and the child comforted the bear, 'Oh poor bear! Are you okay? Do you want to go home?'. The bear agreed and E1 took the bear to her house without feeding her. Finally, E1 explained to the child, 'Oh, you see… the bear was so hurt that she forgot her [name of the object]. We are going to put it here'. E1 removed the food from the box. A second demonstration trial was conducted, and the new item of food was hidden in the other box.

The familiarization phase consisted of four trials (two trials per slide in partially counterbalanced order). The bear first appeared at the door at the bottom of the house, 'Hi, I feel better now! But I am still hungry, so hungry!' E1 offered and hid the piece of food in one box, and then brought a sieve into the scenario, 'Okay. I will put the [name of the object] here in this box for you. So, in order to get the [name of the object] you have to go down the slide, okay? But this time, we have this catching device and [child's name] is going to catch you when you slide!' At the same time, E1 showed how to use it by placing it under each slide. The bear was excited, 'Ok, ok! That's good so I don't get hurt again! Are you ready [child's name]?'. E1 added, addressing the child, 'When the bear is going up the stairs, I will give you the sieve so you can catch the bear, okay?'. The bear then said, 'I'm coming down the slide then! Bye!'. The bear went inside the house and walked up the stairs making walking sounds. E1 pointed to the child, 'Oh the bear is going up the stairs, now it's the time' and gave the sieve to the child. The bear stated 'I am coming! I am coming!' and, after 3 s, she came down the corresponding slide. If the child did not react immediately after holding the sieve, E1 gave a prompt after the 3 s, 'Let's go and catch the bear!' If the child put the sieve under the correct slide, the bear expressed happiness, 'Thank you for catching!' and was fed. If the child held the sieve under the incorrect slide, the bear fell onto the table and got hurt, 'Ouch, ouch'. She trudged back home without eating.

After four familiarization trials, each child performed one test trial in each condition (TB, FB and IG; order counterbalanced). Each trial started with the bear at the door, claiming she was hungry, and E1 offered an item of food. In the TB condition, E1 said, 'Okay, look bear, I put it here in this box for you'. The bear was looking and replied, 'Okay, okay! Mhm, mhm'. Then, another hand puppet (Tom) appeared at the scene. E1 presented Tom to the child, 'Oh, look who is here. This is Tom and look what he is doing'. Tom transferred the food item to the other box while mumbling and left the scene. The bear perceived the transfer and said, 'Ok, then. Bye!'. After saying goodbye to the bear, E1 made the following prompt, 'The bear has seen what Tom did, hasn't he?' and reacted according to the child's answer, 'Yeah, exactly. She has seen it!' (if the child answered correctly) or 'No, she has seen it!' (if the child answered incorrectly). Then, the bear walked up the stairs making walking sounds and E1 handed the sieve over to the child. The bear claimed, 'I am coming! I am coming!'. E1 waited for 3 s and shouted, 'Bear, bear, wait, look what I've found: one of those delicious cupcakes you love so much!! Come downstairs and look!' Hearing this, the bear did not slide. She claimed, 'I will come to the door and have a look!!' and went downstairs.

In the FB condition, everything was exactly the same, but the bear did not witness the transfer. After E1 hid the food in one box, the bear said, 'Ok, then. Bye!' and left the scene. While the bear was absent, Tom appeared and transferred the food. Then, E1 made the prompt, 'The bear hasn't seen what Tom did, has she?' and reacted according to the child's answer, 'Yeah, exactly. She hasn't seen it!' (if the child answered correctly) or 'No, she hasn't seen it!' (if the child answered incorrectly).

In the IG condition, E1 offered the item of food to the bear and said, 'I'll put it in a box'. Before E1 hid the food, the bear claimed, 'Ok, then. Bye!' and left the scene. E1 then stated, 'Okay, I put it in here' and

placed the food in one box. Still in the bear's absence, Tom appeared and transferred the food. E1 said the same prompt as in the FB condition.

Each test condition was followed by a filler trial. As the bear walked down the stairs, E1 removed the previous item of food from the box and the sieve from the child's hands. When the bear appeared at the door, E1 said, 'Look bear, do you like it? I will put it in this box for you'. The bear claimed, 'Okay, okay! Mhm, mhm. Then I go get it!' and left. Inside the house, the bear made walking sounds so E1 handed over the sieve to the child. The bear stated, 'Look out, I am coming! I am coming!' and, after 3 s, she came down the corresponding slide.

### A.2.2. Standard FB task

First, E2 showed the child the boxes, 'Look, I have two play boxes. We can open them and look, there is nothing inside! I put them here'. E2 put the boxes on the floor and then introduced the protagonist and the object, 'Here, there is a dog. And look [child's name], here is the dog's ball. He likes to play with it'. The dog happily played with the object for a few seconds. E2 then stated, 'Oh, but the dog must go home now. But before going home, he puts his ball in this box. And then he leaves'. Meanwhile, E2 placed the object inside one box (order counterbalanced) and took the dog out of the scene. In the dog's absence, E2 introduced a second puppet, 'Look, now the monkey comes. And the monkey takes the ball out the box and puts it in this box. Then, the monkey leaves'. After transferring the ball to the other box, E2 took the monkey out of the scene. E2 then asked the child, 'When the dog comes back, where is he going to look for his ball first?' Three control questions (CQ) followed:

— CQ1: 'Where did the dog put his ball in the beginning?'
— CQ2: 'Where is the ball now?'
— CQ3: 'And did the dog see what the monkey did?'

## A.3. Material

The basic set-up of the interactive change-of-location task consisted of a house made of cardboard (60 cm width × 51 cm height) placed on a table. On each side of the house an opaque, coloured slide (left pink, right yellow) was installed and a same-coloured box (left pink, right yellow, 8 cm tall × 11.5 cm wide × 15.5 cm deep) was placed at the end. At the bottom centre of the house was a black opaque curtain and between the two slides an action camera was installed to record children's looking and catching behaviour from a frontal view. Furthermore, two hand puppets (a bear and a boy), different wooden food items and a sieve were used.

For the standard FB task two stuffed toys (a dog and a monkey), a blue ball and two different coloured boxes (pink and blue, 7 cm tall × 11.4 cm wide × 8.3 cm deep) were used.

## A.4. Coding

A camera was positioned next to the table and captured E1 and the child, and an action camera was arranged behind the house that captured the child's face (in the interactive change-of-location task). Coding was done offline from the video recordings, using the computer software ELAN [60].

In the interactive change-of-location task, children's anticipatory behaviour and anticipatory looking were coded from the time when the sieve was available for the child (after the bear's walking sounds) until E1 presented the more desirable food item. This in total 5 s period included the bear's anticipatory signal (I am coming, I am coming) and the 3 s phase in which children could make a decision.

The differential looking score (DLS) was calculated following the formula:

$$\frac{\text{full slide} - \text{empty slide}}{\text{full slide} + \text{empty slide}}$$

In the standard FB task, we not only coded which box (full or empty) the child referred to after the test question but also after the first two control questions. For the last control question, we coded a yes/ no answer.

We selected 20% of the videos and re-coded them in order to analyse inter-rater reliability. The intraclass correlation coefficient (ICC) for looking and catch behaviours and their 95% confidence

intervals were calculated using R, based on a single-rating, consistency, one-way mixed-effects model. ICC for looking durations was 0.984 (95% CI = 0.98–0.99) and for looking categories was 0.978 (95% CI = 0.97–0.99). ICC for the duration of catch behaviour was 0.977 (95% CI = 0.97–0.99) and for catch categories was 0.963 (95% CI = 0.94–0.98). All ICC values are greater than 0.90 indicating excellent levels of reliability.

# Appendix B. Additional results

## B.1. Familiarization trials

Children's catch performance in each familiarization trial differed from chance level (binomial tests, all $ps < 0.001$). Ninety per cent of participants correctly caught the bear at the end of the full slide in the first familiarization trial, 91% in the second trial, 100% in the third familiarization trials, and 96% did so in the fourth trial.

## B.2. Filler trials

Cochran's Q test indicated that there was not a statistically significant difference between the different filler trials, $\chi_2^2 = 2.235$, $p > 0.05$. Most children correctly caught the bear at the end of the full slide in the TB filler trials (95%), in the FB filler trials (93%) and in the IG filler trials (89%). Children's performance was different from chance level in all filler trials (binomial tests, all $ps < 0.001$).

## B.3. Relation between measures and tasks

Details about the correlations for children's performance in each condition (FB, TB and IG) for each measure (first look, DLS and catch behaviour) with their performance in the standard FB task are shown in tables 2 and 3 (cross-tabulations and correlation coefficients).

## B.4. Results as a function of age group

Figure 3 depicts the main results from the test trials as a function of age group.

## B.5. Results of first trial analyses

We filtered the first trial each child performed in the interactive task and conducted the analyses of the main measures on those trials. Of the final sample ($N = 81$), 30 participants encountered first with the TB condition, 30 with the FB condition and 21 with the IG condition.

### B.5.1. Anticipatory looking

#### B.5.1.1. First look
A chi-square test of independence was performed to examine the relationship between first look and condition. The relation between these variables was significant, $\chi^2(2, N = 80) = 15.5$, $p < 0.001$. *Post hoc* analyses were conducted against a Bonferroni-adjusted alpha level of 0.017 (0.05/3). Children were more likely to look first at the full box in the TB than in the FB condition, $\chi^2(1, N = 60) = 15.0$, $p < 0.001$. However, the proportion of children who looked first at the full box did not differ between TB and IG conditions, $\chi^2(1, N = 50) = 1.59$, $p = .208$, or between FB and IG conditions, $\chi^2(1, N = 50) = 5.56$, $p = 0.018$.

While in the TB condition participants' first look was significantly more often directed to the full box (77%; binomial test, $p = 0.005$), they looked first at the empty box in the FB condition (73%; binomial test, $p = 0.016$). In the IG condition, children performed at chance level (60% looked first at the full box; binomial test, $p = 0.503$).

#### B.5.1.2. Differential looking score
A one-way ANOVA was used to analyse the effect of condition in children's DLS. There was a significant main effect of condition ($F_{2,78} = 5.109$, $p = 0.008$, $\omega^2 = 0.092$).

**Table 2.** Cross-tabulations between children's performance on the standard FB task and first look and catch.

| | | | standard FB task | |
|---|---|---|---|---|
| | | | fail | pass |
| first look | TB | empty box (fail) | 8 | 11 |
| | | full box (pass) | 36 | 26 |
| | FB | empty box (pass) | 20 | 26 |
| | | full box (fail) | 24 | 11 |
| | IG | empty box | 21 | 18 |
| | | full box | 22 | 19 |
| catch | TB | empty box (fail) | 7 | 10 |
| | | full box (pass) | 37 | 26 |
| | FB | empty box (pass) | 22 | 21 |
| | | full box (fail) | 20 | 16 |
| | IG | empty box | 12 | 15 |
| | | full box | 29 | 22 |

**Table 3.** Correlations between children's performance on the standard FB task and the implicit measures.

| | | standard FB task |
|---|---|---|
| first look[a] | TB | 0.136 |
| | FB | 0.25* |
| | IG | 0.002 |
| catch[a] | TB | 0.144 |
| | FB | 0.044 |
| | IG | 0.118 |
| DLS[b] | TB | 0.007 |
| | FB | −0.168 |
| | IG | −0.1 |

[a]Cramer's V.
[b]Point-biserial correlations.
*$p < 0.05$.

*Post hoc* testing using the Bonferroni correction revealed that DLS was significantly lower in the FB condition ($M = -0.32$, s.d. $= 0.70$) compared with the TB condition ($M = 0.29$, s.d. $= 0.80$), $t = -3.089$, $p = .008$, $d = -0.798$. However, DLS was not significantly different between IG ($M = 0.15$, s.d. $= 0.79$) and TB conditions, $t = -0.67$, $p = 0.1$, $d = -0.191$; or between FB and IG conditions, $t = -2.133$, $p = 0.108$, $d = -0.607$.

Separate one-sample *t*-tests indicated that participants preferred to look at the empty box in the FB condition, $t_{29} = -2.478$, $p = 0.019$, $d = -0.452$; but showed no preference to look at one box or the other in the TB condition, $t_{29} = -1.987$, $p = 0.056$, $d = 0.363$, or in the IG condition, $t_{20} = 0.843$, $p = 0.49$, $d = 0.184$.

### B.5.2. Interactive behaviour

A chi-square test of independence was performed to examine the relationship between interactive behaviour and condition. The relation between these variables was significant, $\chi^2(2, N = 78) = 15.6$, $p < 0.001$. *Post hoc* analyses were conducted against a Bonferroni-adjusted alpha level of 0.016 (0.05/3). Children were more likely to catch the bear at the end of the full slide in the TB than in the FB condition, $\chi^2(1, N = 57) = 14.7$, $p < 0.001$; and in the IG condition compared with the FB condition, $\chi^2(1,$

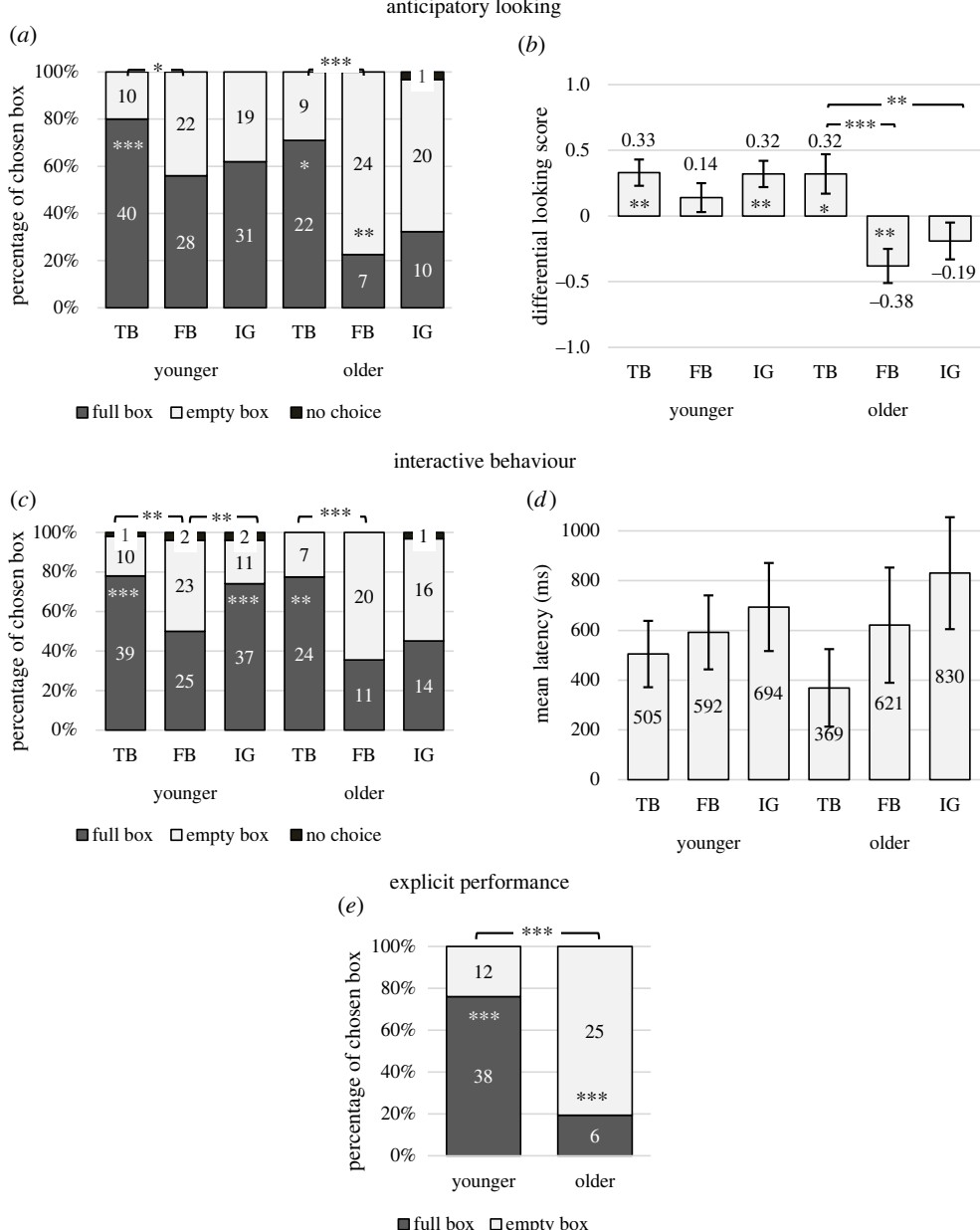

**Figure 3.** Results of the interactive FB task as a function of age group: (*a*) proportion of children's first look at each box (full/empty box), (*b*) differential looking score, (*c*) proportion of children catching the agent under each box (full/empty), (*d*) latency of children's interactive behaviour and (*e*) proportion of children choosing each box (full/box) in the standard FB task. $^*p < 0.05$; $^{**}p < 0.01$; $^{***}p < 0.001$.

$N = 49) = 6.77$, $p = 0.009$. However, the proportion of children who caught the bear at the full slide did not differ between TB and IG conditions, $\chi^2(1, N = 50) = 1.13$, $p = 0.288$.

While in the TB condition children chose to catch the bear significantly more often under the full slide (76%; binomial test, $p = 0.008$), participants' interactive behaviour was directed to the empty slide in the FB condition (75%; binomial test, $p = 0.013$). In IG condition, children performed at chance level (62% chose the full box; binomial test, $p = 0.383$).

### B.5.3. Measures of uncertainty

Kruskal–Wallis H tests showed that there was not a statistically significant difference in latency, $\chi^2_2 = 0.773$, $p = 0.68$; shifts in interactive behaviour, $\chi^2_2 = 0.0004$, $p = 0.98$; and gaze shifts $\chi^2_2 = 5.28$, $p = 0.07$ between the different conditions. Thus, children showed similar latencies as well as shifts in their interactive and looking behaviour in the three conditions.

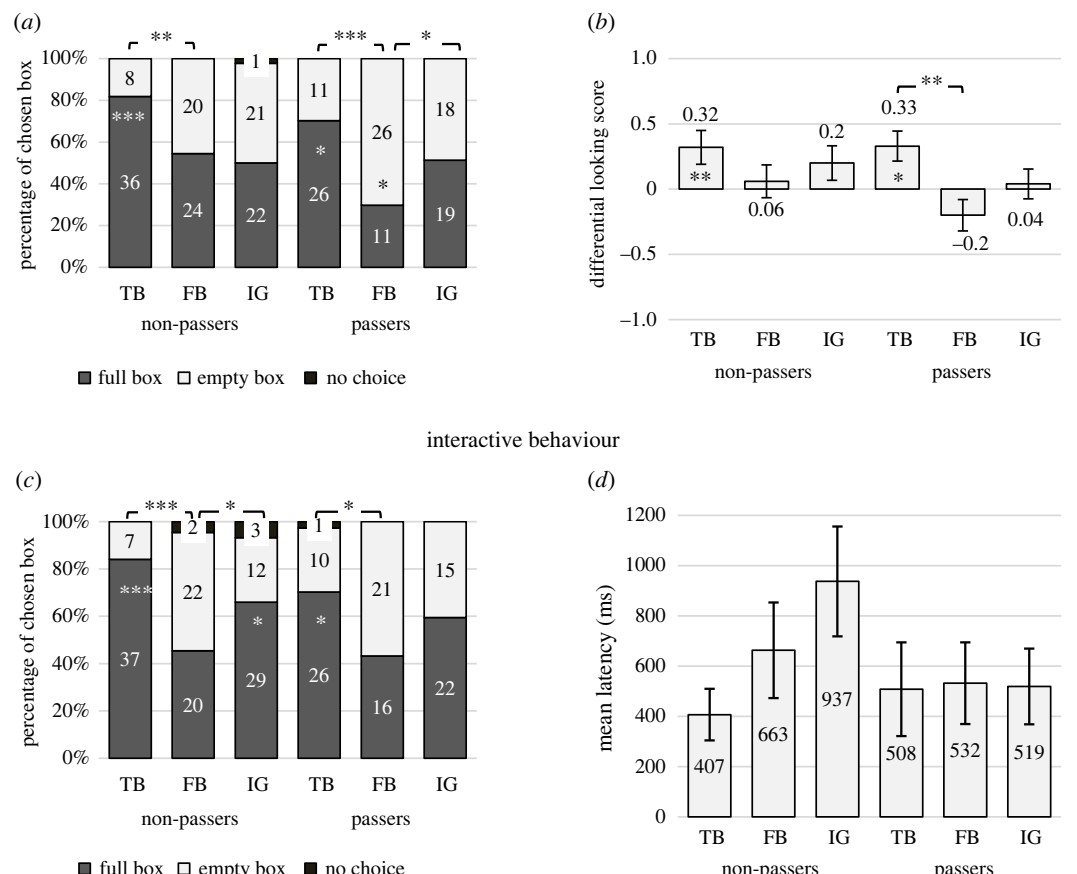

**Figure 4.** Results of the interactive FB task as a function of performance in the standard FB task: (*a*) proportion of children's first look at each box (full/empty box), (*b*) differential looking score, (*c*) proportion of children catching the agent under each slide (full/empty) and (*d*) latency of children's catch behaviour. $^*p < 0.05$; $^{**}p < 0.01$; $^{***}p < 0.001$.

## B.6. Results of standard FB task passers and non-passers

Since one of our main questions concerned children's performance in the interactive task in relation to their ToM abilities, we subdivided the sample according to their performance in the standard FB task. The 'passers' group consisted of 37 children who correctly answered the explicit FB question and predicted that the protagonist will search for her toy in the empty box (18 girls, mean age = 44.3 months). The 'non-passers' group included 44 children who incorrectly chose the full box (17 girls; mean age = 36.86 months). Main results as a function of performance in the standard FB task are depicted in figure 4.

### B.6.1. Anticipatory looking

#### B.6.1.1. First look
A binomial logistic regression was computed in each subgroup (passers and non-passers) to analyse the effect of condition on the likelihood that participants looked first at the full box. FB condition was set as the reference category.

For passers' first look, the logistic regression model was statistically significant, $\chi^2_{108} = 12.537$, $p = 0.002$. Participants were more likely to look first at the full box in the TB condition than in the FB condition ($\beta = 1.72$, $Z_{108} = 3.382$, $p < 0.001$, OR = 5.587). However, children in the IG and FB conditions were as likely to look first at the full box ($\beta = 0.914$, $Z_{108} = 1.876$; $p = 0.061$, OR = 2.495). While in the TB condition passers' first look was directed to the full box significantly more often than chance (70%; binomial test, $p = 0.02$), they looked first at the empty box in the FB condition (70%; binomial test, $p = 0.02$). In the IG condition, passers performed at chance level (51%; binomial test, $p = 1$).

For non-passers, the logistic regression model was statistically significant, $\chi^2_{128} = 11.257$, $p = 0.004$. Participants were more likely to look first at the full box in the TB condition than in the FB condition ($\beta = 1.32$, $Z_{128} = 2.673$, $p = 0.008$, OR = 3.75). However, children in the IG and FB conditions were as likely to look first at the full box ($\beta = -0.136$, $Z(128) = -0.316$; $p = 0.752$, OR = 0.873). Analysed separately, in the TB condition, non-passers looked first at the full box significantly more often than chance (82%; binomial test, $p < 0.001$). In FB and IG conditions children's first look was at chance level (55% and 51% to the full box, respectively; binomial tests, $p > 0.5$).

### B.6.1.2. Differential looking score

A one-way repeated-measures ANOVA was computed in each subgroup (passers versus non-passers) to assess whether there was a difference between conditions in children's DLS.

In the group of standard FB task passers, Mauchly's test of sphericity indicated that the assumption of sphericity had not been violated, $\chi^2_2 = 0.951$, $p = 0.417$. The analysis showed that DLS differed significantly between conditions, $F_{2,72} = 4.936$, $p = 0.01$, $\omega^2 = 0.057$, suggesting that this is a small to medium effect.

*Post hoc* testing using the Bonferroni correction revealed that DLS was significantly higher in TB ($M = 0.33$, s.d. = 0.79) compared with FB condition ($M = -0.20$, s.d. = 0.76), $t_{36} = 3.138$, $p = 0.007$, $d = 0.67$. However, DLS was not significantly different between TB and IG conditions ($M = 0.04$, s.d. = 0.8), $t_{36} = 1.706$, $p = 0.277$, $d = 0.27$; or between FB and IG conditions, $t_{36} = -1.432$, $p = 0.469$, $d = -0.309$.

Separate one-sample *t*-tests indicated that, in the TB condition, passers preferred to look at the full box, $t_{36} = 2.55$, $p = 0.015$, $d = 0.42$; but showed no preference to look at one box or the other in the FB condition, $t_{36} = -1.601$, $p = 0.12$, $d = -0.26$, or in the IG condition, $t_{36} = 0.311$, $p = 0.76$, $d = 0.05$.

In the group of non-passers, Mauchly's test of sphericity indicated that the assumption of sphericity had not been violated, $\chi^2_2 = 0.977$, $p = 0.608$. The analysis showed that DLS did not significantly differ between conditions, $F_{2,86} = 1.241$, $p = 0.294$, $\omega^2 = 0.0004$.

Separate one-sample *t*-tests revealed that, in the TB condition, non-passers preferred to look at the full box, $t_{43} = 2.781$, $p = 0.008$, $d = 0.42$ ($M = 0.32$, s.d. = 0.76). However, they showed no preference to look at one box or the other in the FB condition, $t_{43} = 0.526$, $p = 0.6$, $d = 0.08$ ($M = 0.06$, s.d. = 0.8) or in the IG condition $t_{43} = 1.717$, $p = 0.093$, $d = 0.26$ ($M = 0.2$, s.d. = 0.76).

### B.6.2. Interactive behaviour

A binomial logistic regression was computed in each subgroup (passers and non-passers) to analyse the effect of condition on the likelihood that participants' interactive behaviour was directed to the full box.

For passers, the logistic regression model was statistically significant, $\chi^2_{107} = 6.417$, $p = 0.04$. Participants were more likely to choose the full box in the TB condition ($\beta = 1.227$, $Z_{107} = 2.462$, $p = 0.014$, OR = 3.412) than in the FB condition. However, children in the IG and FB conditions were as likely to choose the full box ($\beta = 0.655$, $Z_{107} = 1.389$; $p = 0.165$, OR = 1.389).

While passers' catch behaviour was significantly more often directed to the full box in the TB condition (72%; binomial test, $p = 0.011$), they chose the full box at chance level in FB and IG conditions (43% and 59%, respectively; binomial tests, $ps > 0.05$).

For non-passers, the logistic regression model was also statistically significant, $\chi^2_{124} = 13.504$, $p = 0.001$. Participants were more likely to choose the full box both in the TB condition ($\beta = 1.76$, $Z_{124} = 3.417$, $p < 0.001$, OR = 5.814) and in the IG condition ($\beta = 0.978$, $Z_{124} = 2.117$, $p < 0.034$, OR = 2.658) than in the FB condition. Non-passers caught the bear significantly more often than chance in the full box in the TB (84%; binomial test, $p < 0.001$) and in the IG conditions (71%; binomial test, $p = 0.012$). In FB condition children chose each box at chance level (48% chose the full box; binomial test, $p = 0.878$).

### B.6.3. Measures of uncertainty

### B.6.3.1. Latency

Friedman's ANOVA showed that latency did not significantly differ among conditions in the passers group, $\chi^2_2 = 2.74$, $p = 0.255$, nor in the non-passers group, $\chi^2_2 = 2.77$, $p = 0.25$. Both groups showed similar latencies in the three conditions (for passers: TB: $M = 508.19$ ms, s.e. = 186.48; FB: $M = 532.11$ ms, s.e. = 162.55; IG: $M = 519.3$ ms, s.e. = 150.7; for non-passers: TB: $M = 406.91$ ms, s.e. = 102.71; FB: $M = 663.295$ ms, s.e. = 190.22; IG: $M = 936.50$ ms, s.e. = 190.23).

### B.6.3.2. Shifts in interactive behaviour

Friedman's ANOVA revealed that shifts in interactive behaviour did not significantly differ among conditions in the passers group, $\chi_2^2 = 1.35$, $p = 0.51$, or in the non-passers group, $\chi_2^2 = 0.448$, $p = 0.799$. Participants of both groups changed their catch behaviour similarly over the three conditions (passers: in the TB condition, range of shifts: 0–1, $M = 0.11$, s.e. = 0.05; in the FB condition, range of shifts: 0–3, $M = 0.19$, s.e. = 0.09; in the IG condition, range of shifts: 0–1, $M = 0.22$, s.e. = 0.069; non-passers: in the TB condition, range of shifts: 0–1, $M = 0.20$, s.e. = 0.06; in the FB condition, range of shifts: 0–1, $M = 0.18$, s.e. = 0.059; in the IG condition, range of shifts: 0–2, $M = 0.30$, s.e. = 0.09).

### B.6.3.3. Shifts in looking behaviour

Friedman's ANOVA revealed that even shifts in looking behaviour did not significantly differ among conditions in the passers group, $\chi_2^2 = 3.41$, $p = 0.181$, or in the non-passers group, $\chi_2^2 = 2.72$, $p = 0.257$. Children of both groups switched their gaze similarly between the two slides in the different conditions (passers: in the TB condition, range of shifts: 0–3, $M = 0.68$, s.e. = 0.15; in the FB condition, range of shifts: 0–4, $M = 1.05$, s.e. = 0.18; in the IG condition, range of shifts: 0–4, $M = 0.95$, s.e. = 0.18; non-passers: in the TB condition, range of shifts: 0–4, $M = 0.64$, s.e. = 0.15; in the FB condition, range of shifts: 0–5, $M = 0.82$, s.e. = 0.18; in the IG condition, range of shifts: 0–3, $M = 1.05$, s.e. = 0.18).

## B.7. Discussion

In general, the results obtained from the appendix roughly portray a similar picture. First trial analyses indicated, on the one hand, that children's looking and interactive behaviours were more likely to be directed to the full box in the TB condition than in the FB condition. On the other hand, children were more likely to choose the full box in the IG condition than in the FB condition in their catch behaviour but not in either of the two looking measures (first look and DLS). Analysing if performance is different from chance in each condition separately, we observed the following pattern in first look and catch measures: TB directed to full box, FB to empty box and IG at chance level. Such a pattern of responses might indeed indicate belief tracking, but only shown in children's first look and interactive behaviour. However, not only is the difference between FB and IG significant in the catch measure, but we should also test more participants conducting the IG condition first to see whether these results hold (as a matter of fact, only 21 children encountered the IG condition first, while 30 children performed either the TB or FB condition first). Future between-subjects studies will shed light on this issue and, in the case they showed converging evidence about belief tracking, we could further discuss the fragility of such a capacity that fades away in a repeated measures study. Moreover, we still found that signs of uncertainty were similar across all conditions.

Secondly, the performance of those children who passed the standard FB task showed that passers failed to differentiate between FB and IG conditions in their DLS and interactive behaviour, although their first look was different in FB and IG conditions. This is another sign consistent with belief tracking but, again, just evident in a single measure (first look). On the other hand, the subsample of children who failed the standard FB task just differentiated FB from IG conditions in their interactive responses, constantly catching the bear at the end of the full slide in the IG condition (not compatible with belief attribution). In neither subsample did children show differences in their signs of uncertainty.

To sum up, if children ascribed proper beliefs, they should have performed in clearly different ways in FB and IG in the different measures we included: looking (first look and DLS), catching, and uncertainty measures (latency, shifts in looking and shifts in catch behaviour). However, they consistently failed to distinguish between these two conditions in five out of the six measures, regardless of the analyses performed (all the sample together with condition as within-subject factor, considering participants' first trials or taking into account their performance on the standard FB task).

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
