## [Peer Review File · Royal Society Open Science]

Review History

RSOS-211278.R0 (Original submission)

Review form: Reviewer 1

Is the manuscript scientifically sound in its present form?

No

Are the interpretations and conclusions justified by the results?

No

Is the language acceptable?

Yes

Do you have any ethical concerns with this paper?

No

Have you any concerns about statistical analyses in this paper?

Yes

Recommendation?

Major revision is needed (please make suggestions in comments)

Comments to the Author(s)

I have read this manuscript with great interest, as it nicely adds to the ongoing debate about the existence and nature of early, implicit mentalizing abilities. The comments I had mainly concern the reporting of the results, as I will outline below.

Quite a few results are described as ‘tending to be significant’ / ‘marginal effects’ / ‘marginally differing’, etc. Such terms can be misleading, so it is best to simply report such results as non-significant. A Bayesian analysis could be computed if the authors wanted to comment on the strength of evidence for either null or alternate model. In addition, the authors often perform follow-up pairwise comparisons even in the absence of significant main effects (and it appears these are not corrected for multiple comparisons). This is not exactly common practice, so unless these comparisons were planned, I would avoid it as much as possible in order not to over-interpret the results.

I was wondering if the authors have any data on the children’s responses to the experimenter’s question about the bear’s perceptual access. This seems a crucial part of the design: whether or not the bear saw what happened, is made explicit to the child. Was there any relation between their initial answer to this question and their performance?

Another question I had was why ‘gender’ was included as a factor in the different models, as this is not really justified. Did the authors have any expectations of gender differences?

Minor comments:

- There are some spelling / language issues, e.g.:

Summary: ‘How robust and reliable are’ -> ‘How robust and reliable is’;

Summary: ‘behaviors towards the agent is measured’ -> ‘behaviors towards the agent are measured’;

Methods (3.3): ‘look one side over the other’ -> ‘look at one side over the other’;

Appendix A (3): ‘made of cupboard’ -> ‘made of cartboard’?

Please check for such mistakes throughout.

- Methods, 3.1: ‘Around their 4th birthday is still relatively young, not all children may have developed explicit ToM abilities fully. Can the authors comment on how this may have influenced results?’

- Results, 4.1.1: The description of the order effects was not quite clear to me. Did they look more at the full box on all three conditions for ‘IG-TB-FB’ order, or on one/two conditions in particular?

- Results, 4.5 has the same title as 4.4 – this might have to be rephrased?

- Figure 2: the stripes indicating which conditions significantly differ from one another seem to be misplaced.

- Appendix A, ‘2. Participants’ -> this seems to be the wrong title.

- Appendix B (2): Were the differences between the different filler trials significant?

(Worse performance after IG condition?)

Review form: Reviewer 2

Is the manuscript scientifically sound in its present form?

Yes

Are the interpretations and conclusions justified by the results?

No

Is the language acceptable?

Yes

Do you have any ethical concerns with this paper?

No

Have you any concerns about statistical analyses in this paper?

No

Recommendation?

Major revision is needed (please make suggestions in comments)

Comments to the Author(s)

The present study is a conceptual replication of a combined interactive-spontaneous and verbal-theory of mind task, based on Garnham and Perner (2001). The goals of the study are ambitious: probe the replicability of the paradigm, extend it, and address the theoretical debates surrounding the interpretation of children's behavior in the task. This is an important endeavor, and in general I very much support replication projects to be published. In the present form, however, I find that a more balanced picture of the literature as well as a more detailed and nuanced discussion of the current findings should be presented. Below I list some points to address.

Introduction:

1. The original study is not presented in the introduction, and thus is not clear in what sense this is a replication: what paradigm and setup did the original include, and what exactly did they analyze and find.
2. This should/could be done instead of the rather general introduction of various implicit ToM paradigms, e.g. VoE is not strictly relevant for the current paper, whereas e.g. not much is mentioned about the interactive tasks. (in addition, Kiraly et al, PNAS, is mentioned as a non-replication but if I remember well they replicated the original findings).
3. Relatedly, some of the literature would be very relevant to introduce, on tasks that are similar to the Garnham and Perner one, e.g. studies by Scott et al, other interactive tasks, or paradigms that for example have spontaneous measures to verbal stories.
4. There is also a large literature on the verbal ToM tasks and their critique, stability of passing within an individual child, papers that disagree with the fact that verbal FB tasks should measure metarepresentations or that it should be used as a litmus test. There are also studies showing that the capacities involved in these tasks may develop further after age 4. These should be at least mentioned, especially since the current study does not probe metarepresentations directly, and since debates around the implicit measures are discussed in some detail. It is not the case that the verbal or "standard" FB tasks stand without critique and I would be helpful not gloss over these arguments, especially when looking at the relation of this task to other measures, where it may be relevant to think about what capacities may be involved in the different tasks. If not in the introduction, then in the discussion considering these would be informative.

5. Regarding the theoretical claims in the introduction: if the authors aim to include debates on metarepresentations, it would be relevant to explain which implicit tasks are proposed to measure these and why, as currently it is simply stated (cf. p. 3 line 36-40).

6. In order to situate it in the (non)-replication literature and findings, it could be relevant to mention, why did the authors choose this particular paradigm, what does it add to previous ones, and how does it differ from the other (non-) replications?

Results:

7. In general, in order to probe these various different effects, this study may be underpowered. This is also suggested by the many non-significant “trend” effects (which are generally advised not to be used as a term, as in frequentist statistics it is somewhat unclear what it means), which may come from the lack of power. Bayesian statistics could also help (though not solve) the absence of effects, where instead the confidence in the null could be expressed. I am aware of the power calculations the authors did before the study, but that did not include the ignorance condition or the hesitance measures. In fact, it seems to me that those effects that were based on the power calculations largely replicated, but I may be mistaken.

8. For the first look (and perhaps for the other measures too) it would be useful to do a first trial analysis. This is also shown by the interaction between order and first look, which is hard to interpret, and the authors do not return to. If there is not enough power in the current sample, then perhaps in a follow-up to strengthen and clarify these results.

9. In the first look, it is mentioned that there is a significant difference between FB and IG, which if true, then the claim in the discussion (p. 9, line 9) is not correct and there are differences between FB and IG. But it seems to me this may be a typo?

10. In DLS, $p = .087$ is referred to as tending to significance but $p = .11$ is not (former is the difference between TB and IG, latter between FB and IG), I am sympathetic to not only considering effects below the $p = .05$ cutoff, but then some other way of discussing/analyzing the results should be added (e.g. Bayesian statistics).

11. In catching behavior, children in IG chose the full box above chance, not consistently with the seeing=knowing rule that is proposed.

12. While in the latency, catch-shifts, and look-shifts measures the effects are generally not significant, all three follow the TB < FB < IG order. Perhaps the two shifts could be merged into a composite measure, but in any case, this is a pattern that may be worth commenting on. I am not convinced that this shows that there was equal amount of hesitance in FB and IG. In addition, neither of these 3 hesitance measures show that TB and FB differ from each other, so should we then conclude that children equated TB and FB?

13. Passers in first look appear to me to behave consistently with belief ascription and not seeing = knowing: above-chance to full box in TB, above-chance to empty in FB, and chance in IG. Non-passers on the other hand look at chance in FB. This pattern should be commented on. (similarly in DLS, when considering younger vs. older kids instead of passing on explicit task, in the older group the entire pattern seems to me consistent with belief attribution). Relatedly, the authors report that first look in the FB condition was associated with the explicit FB performance. Of the things that could be related between the two tasks, this is arguably one of the most relevant to probe whether the interactive and explicit measures are related.

Discussion:

14. Overall, it seems to me that the ignorance condition may not have worked as intended. It may stem from the implementation: the interaction between E1 and the bear may indicate that the bear will know where she puts it, and then when Tom relocates it, it effectively turns into a false belief. But in fact, in some of the measures, children act as if the bear had a true belief (e.g., in catching behavior, and especially the younger children). This potentially confusing nature of the IG condition may be reflected in the hesitance measures, where on all three the IG condition shows most hesitance. This seems especially true in non-passers.

15. Is it possible that the question: “The bear hasn’t seen what Tom did there, has he?” prompted children to use a seeing-knowing strategy? This would also make the FB and IG conditions effectively similar to each other.
16. On p. 8 line 55-56 it is stated that in first look children showed a tendency to look first to the full box in IG, but from what I gather from the results, this was not the case.
17. Throughout, it is suggested that the verbal/standard FB task is harder than the interactive ones, because the latter involves implicit ToM which is simpler/easier. But in fact we cannot be sure that the interactive part involves “implicit” processing especially at this age; and if we consider that some (or all) of these children may undergo some cognitive changes related to the verbal ToM (whatever they may be, unrelated to exact theoretical commitment) then they may be reasoning explicitly and then have to plan their actions based on that, which may make the interactive part more difficult. A more elaborate discussion on the measures could benefit the paper.
18. I am a little confused about the conclusions. The interactive measures are robust and increase with age, but they measure a simpler mechanism - so this simpler mechanism increases with age similarly to the metarepresentational capacities? It may be informative to relate this to other literature on knowledge-ignorance or seeing=knowing.
19. Relatedly, how does one explain those who passed the explicit FB and didn’t pass the implicit ones: they have metarepresentations but cannot apply these when they need to catch someone?
20. The ‘seeing leads to knowing mechanism could be elaborated on. For example, why doesn’t not seeing lead to being mistaken? How do children represent others’ ignorance?
21. On p. 8, line 57 it is mentioned that the various measures in the interactive task were related to each other. I apologize if I missed this, but I did not find the analyses this is based on.

Overall, the results seem mixed and while some results replicate, others don’t (e.g., I think in the original study the looks were significantly above chance in the belief-direction also in FB); and these should be detailed and discussed. Additionally, in order to make larger theoretical claims, a more fine-grained and critical discussion of the results should be included, with those patterns also acknowledged that may warrant more nuanced interpretations.

Currently, it does not seem to me that it is entirely clear what the data show related to the three targeted questions. To answer question (1) a description of the original findings would be needed, and discussing the current ones in more detail; regarding (2) it appears that the manipulation may not have worked ideally, yet in some measures there may be some differences between FB and IG, whereas in contrast in others the pattern of response appears not consistent with a seeing=knowing rule, and for point (3) I could not find where the relation between interactive measures are reported (again, apologies if I missed it), and in some measures there is a relation with the explicit task, or response pattern consistent with belief attribution in the implicit tasks, specifically in explicit-passers only. Together, addressing these would contribute greatly to the existing literature and move towards clarifying replicability of ToM paradigms and cognitive foundations of ToM.

Smaller remarks

22. Figure 2, especially Fig 2D, appears somewhat scrambled and the error bars (?) are shifted. Or if that’s not the case then I am not sure how to interpret the horizontal lines across the figures.
23. In supplemental Figure 1 the labels for younger children appear somewhat scrambled.
24. Sections 4.4 and 4.5 have the same name (“Standard FB task”)

Decision letter (RSOS-211278.R0)

Dear Dr Barone

The Editors assigned to your paper RSOS-211278 "Do Young Children Track Other's Beliefs, or Merely their Perceptual Access? An Interactive, Anticipatory Measure of Early Theory of Mind" have now received comments from reviewers and would like you to revise the paper in accordance with the reviewer comments and any comments from the Editors. Please note this decision does not guarantee eventual acceptance.

Please submit your revised manuscript and required files (see below) no later than 21 days from today's (ie 09-Nov-2021) date. Note: the ScholarOne system will 'lock' if submission of the revision is attempted 21 or more days after the deadline. If you do not think you will be able to meet this deadline please contact the editorial office immediately.

on behalf of Dr Carina de Klerk (Associate Editor) and Essi Viding (Subject Editor)
openscience@royalsociety.org

Associate Editor Comments to Author (Dr Carina de Klerk):
Comments to the Author:

As you will see from the enclosed comments both reviewers find the work interesting and believe that it adds to the ongoing debate about early theory of mind development. However, both reviewers also raised a number of important issues that need to be addressed. In particular, both reviewers commented on the way the results are described, suggesting you should refrain from

using terms like ‘marginal’ to describe non-significant findings and both suggest you could use Bayesian analyses to clarify null results. Furthermore, reviewer 2 notes that a more balanced representation of the literature should be presented in the introduction and discussion section. This reviewer additionally points out several important issues with the interpretation of the findings, in particular relating to the ignorance condition and the strength of the evidence for the seeing=knowing rule, which will need to be addressed in a revision.

Reviewer comments to Author:

Reviewer: 1

Comments to the Author(s)

I have read this manuscript with great interest, as it nicely adds to the ongoing debate about the existence and nature of early, implicit mentalizing abilities. The comments I had mainly concern the reporting of the results, as I will outline below.

Quite a few results are described as ‘tending to be significant’ / ‘marginal effects’ / ‘marginally differing’, etc. Such terms can be misleading, so it is best to simply report such results as non-significant. A Bayesian analysis could be computed if the authors wanted to comment on the strength of evidence for either null or alternate model. In addition, the authors often perform follow-up pairwise comparisons even in the absence of significant main effects (and it appears these are not corrected for multiple comparisons). This is not exactly common practice, so unless these comparisons were planned, I would avoid it as much as possible in order not to over-interpret the results.

I was wondering if the authors have any data on the children’s responses to the experimenter’s question about the bear’s perceptual access. This seems a crucial part of the design: whether or not the bear saw what happened, is made explicit to the child. Was there any relation between their initial answer to this question and their performance?

Another question I had was why ‘gender’ was included as a factor in the different models, as this is not really justified. Did the authors have any expectations of gender differences?

Minor comments:

- There are some spelling / language issues, e.g.:

Summary: ‘How robust and reliable are’ -> ‘How robust and reliable is’;

Summary: ‘behaviors towards the agent is measured’ -> ‘behaviors towards the agent are measured’;

Methods (3.3): ‘look one side over the other’ -> ‘look at one side over the other’;

Appendix A (3): ‘made of cupboard’ -> ‘made of cartboard’?

Please check for such mistakes throughout.

- Methods, 3.1: ‘Around their 4th birthday is still relatively young, not all children may have developed explicit ToM abilities fully. Can the authors comment on how this may have influenced results?’

- Results, 4.1.1: The description of the order effects was not quite clear to me. Did they look more at the full box on all three conditions for ‘IG-TB-FB’ order, or on one/two conditions in particular?

- Results, 4.5 has the same title as 4.4 – this might have to be rephrased?

- Figure 2: the stripes indicating which conditions significantly differ from one another seem to be misplaced.

- Appendix A, ‘2. Participants’ -> this seems to be the wrong title.

- Appendix B (2): Were the differences between the different filler trials significant? (Worse performance after IG condition?)

Reviewer: 2

Comments to the Author(s)

The present study is a conceptual replication of a combined interactive-spontaneous and verbal-theory of mind task, based on Garnham and Perner (2001). The goals of the study are ambitious: probe the replicability of the paradigm, extend it, and address the theoretical debates surrounding the interpretation of children's behavior in the task. This is an important endeavor, and in general I very much support replication projects to be published. In the present form, however, I find that a more balanced picture of the literature as well as a more detailed and nuanced discussion of the current findings should be presented. Below I list some points to address.

Introduction:

1. The original study is not presented in the introduction, and thus is not clear in what sense this is a replication: what paradigm and setup did the original include, and what exactly did they analyze and find.
2. This should/could be done instead of the rather general introduction of various implicit ToM paradigms, e.g. VoE is not strictly relevant for the current paper, whereas e.g. not much is mentioned about the interactive tasks. (in addition, Kiraly et al, PNAS, is mentioned as a non-replication but if I remember well they replicated the original findings).
3. Relatedly, some of the literature would be very relevant to introduce, on tasks that are similar to the Garnham and Perner one, e.g. studies by Scott et al, other interactive tasks, or paradigms that for example have spontaneous measures to verbal stories.
4. There is also a large literature on the verbal ToM tasks and their critique, stability of passing within an individual child, papers that disagree with the fact that verbal FB tasks should measure metarepresentations or that it should be used as a litmus test. There are also studies showing that the capacities involved in these tasks may develop further after age 4. These should be at least mentioned, especially since the current study does not probe metarepresentations directly, and since debates around the implicit measures are discussed in some detail. It is not the case that the verbal or "standard" FB tasks stand without critique and I would be helpful not gloss over these arguments, especially when looking at the relation of this task to other measures, where it may be relevant to think about what capacities may be involved in the different tasks. If not in the introduction, then in the discussion considering these would be informative.
5. Regarding the theoretical claims in the introduction: if the authors aim to include debates on metarepresentations, it would be relevant to explain which implicit tasks are proposed to measure these and why, as currently it is simply stated (cf. p. 3 line 36-40).
6. In order to situate it in the (non)-replication literature and findings, it could be relevant to mention, why did the authors choose this particular paradigm, what does it add to previous ones, and how does it differ from the other (non-) replications?

Results:

7. In general, in order to probe these various different effects, this study may be underpowered. This is also suggested by the many non-significant "trend" effects (which are generally advised not to be used as a term, as in frequentist statistics it is somewhat unclear what it means), which may come from the lack of power. Bayesian statistics could also help (though not solve) the absence of effects, where instead the confidence in the null could be expressed. I am aware of the power calculations the authors did before the study, but that did not include the ignorance condition or the hesitance measures. In fact, it seems to me that those effects that were based on the power calculations largely replicated, but I may be mistaken.
8. For the first look (and perhaps for the other measures too) it would be useful to do a first trial analysis. This is also shown by the interaction between order and first look, which is hard to interpret, and the authors do not return to. If there is not enough power in the current sample, then perhaps in a follow-up to strengthen and clarify these results.

9. In the first look, it is mentioned that there is a significant difference between FB and IG, which if true, then the claim in the discussion (p. 9, line 9) is not correct and there are differences between FB and IG. But it seems to me this may be a typo?
10. In DLS, $p = .087$ is referred to as tending to significance but $p = .11$ is not (former is the difference between TB and IG, latter between FB and IG), I am sympathetic to not only considering effects below the $p = .05$ cutoff, but then some other way of discussing/analyzing the results should be added (e.g. Bayesian statistics).
11. In catching behavior, children in IG chose the full box above chance, not consistently with the seeing=knowing rule that is proposed.
12. While in the latency, catch-shifts, and look-shifts measures the effects are generally not significant, all three follow the $TB < FB < IG$ order. Perhaps the two shifts could be merged into a composite measure, but in any case, this is a pattern that may be worth commenting on. I am not convinced that this shows that there was equal amount of hesitance in FB and IG. In addition, neither of these 3 hesitance measures show that TB and FB differ from each other, so should we then conclude that children equated TB and FB?
13. Passers in first look appear to me to behave consistently with belief ascription and not seeing = knowing: above-chance to full box in TB, above-chance to empty in FB, and chance in IG. Non-passers on the other hand look at chance in FB. This pattern should be commented on. (similarly in DLS, when considering younger vs. older kids instead of passing on explicit task, in the older group the entire pattern seems to me consistent with belief attribution). Relatedly, the authors report that first look in the FB condition was associated with the explicit FB performance. Of the things that could be related between the two tasks, this is arguably one the most relevant to probe whether the interactive and explicit measures are related.

Discussion:

14. Overall, it seems to me that the ignorance condition may not have worked as intended. It may stem from the implementation: the interaction between E1 and the bear may indicate that the bear will know where she puts it, and then when Tom relocates it, it effectively turns into a false belief. But in fact, in some of the measures, children act as if the bear had a true belief (e.g., in catching behavior, and especially the younger children). This potentially confusing nature of the IG condition may be reflected in the hesitance measures, where on all three the IG condition shows most hesitance. This seems especially true in non-passers.
15. Is it possible that the question: "The bear hasn't seen what Tom did there, has he?" prompted children to use a seeing-knowing strategy? This would also make the FB and IG conditions effectively similar to each other.
16. On p. 8 line 55-56 it is stated that in first look children showed a tendency to look first to the full box in IG, but from what I gather from the results, this was not the case.
17. Throughout, it is suggested that the verbal/standard FB task is harder than the interactive ones, because the latter involves implicit ToM which is simpler/easier. But in fact we cannot be sure that the interactive part involves "implicit" processing especially at this age; and if we consider that some (or all) of these children may undergo some cognitive changes related to the verbal ToM (whatever they may be, unrelated to exact theoretical commitment) then they may be reasoning explicitly and then have to plan their actions based on that, which may make the interactive part more difficult. A more elaborate discussion on the measures could benefit the paper.
18. I am a little confused about the conclusions. The interactive measures are robust and increase with age, but they measure a simpler mechanism - so this simpler mechanism increases with age similarly to the metarepresentational capacities? It may be informative to relate this to other literature on knowledge-ignorance or seeing=knowing.
19. Relatedly, how does one explain those who passed the explicit FB and didn't pass the implicit ones: they have metarepresentations but cannot apply these when they need to catch someone?
20. The 'seeing leads to knowing mechanism could be elaborated on. For example, why doesn't not seeing lead to being mistaken? How do children represent others' ignorance?

21. On p. 8, line 57 it is mentioned that the various measures in the interactive task were related to each other. I apologize if I missed this, but I did not find the analyses this is based on.

Overall, the results seem mixed and while some results replicate, others don't (e.g., I think in the original study the looks were significantly above chance in the belief-direction also in FB); and these should be detailed and discussed. Additionally, in order to make larger theoretical claims, a more fine-grained and critical discussion of the results should be included, with those patterns also acknowledged that may warrant more nuanced interpretations.

Currently, it does not seem to me that it is entirely clear what the data show related to the three targeted questions. To answer question (1) a description of the original findings would be needed, and discussing the current ones in more detail; regarding (2) it appears that the manipulation may not have worked ideally, yet in some measures there may be some differences between FB and IG, whereas in contrast in others the pattern of response appears not consistent with a seeing=knowing rule, and for point (3) I could not find where the relation between interactive measures are reported (again, apologies if I missed it), and in some measures there is a relation with the explicit task, or response pattern consistent with belief attribution in the implicit tasks, specifically in explicit-passers only. Together, addressing these would contribute greatly to the existing literature and move towards clarifying replicability of ToM paradigms and cognitive foundations of ToM.

Smaller remarks

22. Figure 2, especially Fig 2D, appears somewhat scrambled and the error bars (?) are shifted. Or if that's not the case then I am not sure how to interpret the horizontal lines across the figures.

23. In supplemental Figure 1 the labels for younger children appear somewhat scrambled.

24. Sections 4.4 and 4.5 have the same name ("Standard FB task")

===PREPARING YOUR MANUSCRIPT===

If you have been asked to revise the written English in your submission as a condition of publication, you must do so, and you are expected to provide evidence that you have received language editing support. The journal would prefer that you use a professional language editing service and provide a certificate of editing, but a signed letter from a colleague who is a fluent

speaker of English is acceptable. Note the journal has arranged a number of discounts for authors using professional language editing services (<https://royalsociety.org/journals/authors/benefits/language-editing/>).

===PREPARING YOUR REVISION IN SCHOLARONE===

<https://royalsociety.org/journals/authors/author-guidelines/#supplementary-material> to include a suitable title and informative caption. An example of appropriate titling and captioning may be found at https://figshare.com/articles/Table_S2_from_Is_there_a_trade-

off_between_peak_performance_and_performance_breadth_across_temperatures_for_aerobic_sc
ope_in_teleost_fishes_/3843624.

Author's Response to Decision Letter for (RSOS-211278.R0)

See Appendix A.

RSOS-211278.R1 (Revision)

Review form: Reviewer 2

Is the manuscript scientifically sound in its present form?

Yes

Are the interpretations and conclusions justified by the results?

No

Is the language acceptable?

Yes

Do you have any ethical concerns with this paper?

No

Have you any concerns about statistical analyses in this paper?

No

Recommendation?

Major revision is needed (please make suggestions in comments)

Comments to the Author(s)

Generally, the manuscript has improved substantially, is more clear, and I applaud the changes the authors made, and appreciate most of their responses. I have one broad, major issue still, which is that I find the ignorance condition, as I pointed out at various points, not convincing. It could be partially the implementation, and largely sample size, and potentially other factors. If there was a way to show that this sample size was big enough to find these effects then I this argument would not hold as much; but currently there is no such evidence provided, and many (if not most) of the reported effects are in the right direction just not significant. I believe that while the IG condition was a creative addition that could yield strong results, we cannot be sure if it succeeded. As such, I find the point with the replication fully convincing, the relation with the explicit task moderately (as 1 out of 3 measures show a relation); and the alternative theoretical explanation not convincing, and suggest to revise the abstract and discussion accordingly, and provide a balanced presentation of all results (e.g. if 1 of 3 effects is significant then instead of "largely not significant" – simply the number can be described).

I also have several smaller points which I outline below (some are related to my above point).

1) Now I understand better the rationale of the present study, but the abstract and the intro do not reflect this, except the last sentences of the introduction. While replication is generally a reasonable endeavor, there are also many tasks out there by now. I strongly believe that it is necessary to justify and define from the get-go what the scope and goal here is.

2) My point with raising the power analysis (point 7 in my previous review) is not simply to add a comment in the methods that IG condition was not added to the power analysis. I think it is important to mention as limitation and dampen conclusions throughout which refer to effects that may be underpowered. The authors should add this in the main text and add a section discussing that some of the contrasts may have depended on this. As I mentioned, to me it looks that those effects that were sufficiently powered based on the power calculation, did replicate, in line with the possibility that not sufficient power may be an issue in the others. Perhaps post-hoc power analyses could be performed for the critical comparisons.

3) In point 11 in my previous review I pointed out that children in the IG chose the full box above chance which is not consistent with the proposed rule. While it may be that the crucial contrast for the authors is FB vs. IG, but if they wish to make theoretical claims regarding what cognitive mechanisms may underlie children's behavior then these should be consistent with the observed pattern.

4) Why did the authors delete the Wilcoxon tests? (I am not criticizing, merely wondering)

5) I agree with Rev 1 that gender is unnecessary to add, unless there is a hypothesis for it

6) I understand now that the results on the significant correlations within interactive measures were not included, and now understand what the discussion referred to. However, since this has been also subject to much debate, I would find it important to comment on the fact that within the implicit measures the correlations were significant; definitely in the discussion, but perhaps even in the abstract; especially considering that the authors mention this as one of the goals of the present paper / using this task

7) The authors not find the effect by Rakoczy et al re: FB-passers start failing TB - is it perhaps relevant to comment on?

8) The discussion is more balanced now, but still not completely. E.g., on p.37, lines 48-49 the authors write that children's performance should differ drastically between FB and IG. This is not really the case, as in IG we'd expect chance and in FB an effect, but this may need sufficient power to reach as it is not an opposite direction of effect expected (which is in FB vs TB). (This would be a good place to mention the issue of power.) There is also a lot of the discussion addressed to the FB vs IG, arguing still, and with increasingly strong wording, for alternative mechanisms. This on one hand is repetitive, and on the other hand there are other aspects of the results to mention while this is toned down- currently it is worded more strongly than what I'd say the results warrant.

9) Currently in the appendix B, 5.1.1 first look It says FB vs. IG $p=0.018$, which would be significant; and their pattern is consistent with belief ascription not seeing = knowing rule; and not consistently with that the discussion in the appendix says.

10) In my previous point 12, according to common notation, I used TB < FB < IG to express there was more catch-shift in FB than in TB, and more in IG than in FB (TB < FB refers to FB having more than TB). To me it still looks like it is TB < FB < IG in all three, e.g. latency in TB is 453.17ms, in FB 603.37ms, and in IG 745.93; the catch-shift in TB=.16, FB=.19, and in IG=.26, suggesting more catch-shift in IG; and similarly in look-shifts TB=.65, FB=.93, and IG=1; again corresponding to the same order. If I am reading these wrong I would appreciate if the authors explained the misunderstanding by referring to the numbers. I understand that (some/all? of) these are not significant but as again this may be a power issue, at least acknowledging this pattern would be informative for the reader, as there are many results described and it can get lost in the mix; and this order of conditions, as far as I understand, is the one predicted by belief attribution.

- 11) In the first paragraph of the discussion the questions and answers the second two are reversed (the third answer replies Q2 and vice versa)
- 12) P. 38 lines 16 and 24 both start with "all in all"
- 13) The discussion from the appendix should be moved to the main text. If word limit prevents, some of the repetitive parts of the discussion can be streamlined (several of the paragraphs discuss the FB vs. IG / the seeing = knowing rule; factive mindreading, etc.
- 14) To my reply to point 15, the authors mention that in some measures FB and IG differed; to argue against the possibility that the test question biased children. However, this argument supposes that we take this finding seriously; which would mean that the findings overall are not consistent with the alternative rule. This is an inconsistency to be addressed – either much dampen the emphasis on the alternative rule, or explicitly acknowledge that this way of phrasing the question may in fact have elicited interpretations based on this rule (since the prompt is pointing out the fact that the bear hasn't seen what Tom did)
- 15) Cf. my previous point 18, I still don't understand why, if children use a simpler mechanism that explains young children and infants' abilities such as seeing = knowing, there is an age effect in the implicit measures. Such rules are typically meant to explain data of infants as young as 6 months of age; why, then, is there an increase between 3 to 4 years?

Decision letter (RSOS-211278.R1)

Dear Dr Barone

The Editors assigned to your paper RSOS-211278.R1 "Do Young Children Track Other's Beliefs, or Merely their Perceptual Access? An Interactive, Anticipatory Measure of Early Theory of Mind" have now received comments from reviewers and would like you to revise the paper in accordance with the reviewer comments and any comments from the Editors. Please note this decision does not guarantee eventual acceptance.

Please submit your revised manuscript and required files (see below) no later than 21 days from today's (ie 27-May-2022) date. Note: the ScholarOne system will 'lock' if submission of the revision is attempted 21 or more days after the deadline. If you do not think you will be able to meet this deadline please contact the editorial office immediately.

Please note article processing charges apply to papers accepted for publication in Royal Society Open Science (<https://royalsocietypublishing.org/rsos/charges>). Charges will also apply to papers transferred to the journal from other Royal Society Publishing journals, as well as papers submitted as part of our collaboration with the Royal Society of Chemistry

(<https://royalsocietypublishing.org/rsos/chemistry>). Fee waivers are available but must be requested when you submit your revision (<https://royalsocietypublishing.org/rsos/waivers>).

on behalf of Dr Carina de Klerk (Associate Editor) and Essi Viding (Subject Editor)
openscience@royalsociety.org

Associate Editor Comments to Author (Dr Carina de Klerk):

Associate Editor: 1

Comments to the Author:

Dear authors,

The revised manuscript was sent back to one of the original reviewers and although they indicate that the manuscript is much improved, they also identified several remaining issues that would need to be addressed before the manuscript is suitable for publication. Although Royal Society Open Science does not normally allow for multiple rounds of revision, the Subject Editor and I believe it should be possible to address the remaining issues in one final round of revision and we therefore invite you to revise and resubmit your manuscript. Please make sure you address all the reviewer's remaining comments in your revision.

Reviewer comments to Author:

Reviewer: 2

Comments to the Author(s)

Generally, the manuscript has improved substantially, is more clear, and I applaud the changes the authors made, and appreciate most of their responses. I have one broad, major issue still, which is that I find the ignorance condition, as I pointed out at various points, not convincing. It could be partially the implementation, and largely sample size, and potentially other factors. If there was a way to show that this sample size was big enough to find these effects then I this argument would not hold as much; but currently there is no such evidence provided, and many (if not most) of the reported effects are in the right direction just not significant. I believe that while the IG condition was a creative addition that could yield strong results, we cannot be sure if it succeeded. As such, I find the point with the replication fully convincing, the relation with the explicit task moderately (as 1 out of 3 measures show a relation); and the alternative theoretical explanation not convincing, and suggest to revise the abstract and discussion accordingly, and provide a balanced presentation of all results (e.g. if 1 of 3 effects is significant then instead of "largely not significant" – simply the number can be described).

I also have several smaller points which I outline below (some are related to my above point).

- 1) Now I understand better the rationale of the present study, but the abstract and the intro do not reflect this, except the last sentences of the introduction. While replication is generally a reasonable endeavor, there are also many tasks out there by now. I strongly believe that it is necessary to justify and define from the get-go what the scope and goal here is.
- 2) My point with raising the power analysis (point 7 in my previous review) is not simply to add a comment in the methods that IG condition was not added to the power analysis. I think it is important to mention as limitation and dampen conclusions throughout which refer to effects

that may be underpowered. The authors should add this in the main text and add a section discussing that some of the contrasts may have depended on this. As I mentioned, to me it looks that those effects that were sufficiently powered based on the power calculation, did replicate, in line with the possibility that not sufficient power may be an issue in the others. Perhaps post-hoc power analyses could be performed for the critical comparisons.

3) In point 11 in my previous review I pointed out that children in the IG chose the full box above chance which is not consistent with the proposed rule. While it may be that the crucial contrast for the authors is FB vs. IG, but if they wish to make theoretical claims regarding what cognitive mechanisms may underlie children's behavior then these should be consistent with the observed pattern.

4) Why did the authors delete the Wilcoxon tests? (I am not criticizing, merely wondering)

5) I agree with Rev 1 that gender is unnecessary to add, unless there is a hypothesis for it

6) I understand now that the results on the significant correlations within interactive measures were not included, and now understand what the discussion referred to. However, since this has been also subject to much debate, I would find it important to comment on the fact that within the implicit measures the correlations were significant; definitely in the discussion, but perhaps even in the abstract; especially considering that the authors mention this as one of the goals of the present paper / using this task

7) The authors not find the effect by Rakoczy et al re: FB-passers start failing TB – is it perhaps relevant to comment on?

8) The discussion is more balanced now, but still not completely. E.g., on p.37, lines 48-49 the authors write that children's performance should differ drastically between FB and IG. This is not really the case, as in IG we'd expect chance and in FB an effect, but this may need sufficient power to reach as it is not an opposite direction of effect expected (which is in FB vs TB). (This would be a good place to mention the issue of power.) There is also a lot of the discussion addressed to the FB vs IG, arguing still, and with increasingly strong wording, for alternative mechanisms. This on one hand is repetitive, and on the other hand there are other aspects of the results to mention while this is toned down- currently it is worded more strongly than what I'd say the results warrant.

9) Currently in the appendix B, 5.1.1 first look It says FB vs. IG $p=0.018$, which would be significant; and their pattern is consistent with belief ascription not seeing = knowing rule; and not consistently with that the discussion in the appendix says.

10) In my previous point 12, according to common notation, I used $TB < FB < IG$ to express there was more catch-shift in FB than in TB, and more in IG than in FB ($TB < FB$ refers to FB having more than TB). To me it still looks like it is $TB < FB < IG$ in all three, e.g. latency in TB is 453.17ms, in FB 603.37ms, and in IG 745.93; the catch-shift in $TB=.16$, $FB=.19$, and in $IG=.26$, suggesting more catch-shift in IG; and similarly in look-shifts $TB=.65$, $FB=.93$, and $IG=1$; again corresponding to the same order. If I am reading these wrong I would appreciate if the authors explained the misunderstanding by referring to the numbers. I understand that (some/all? of) these are not significant but as again this may be a power issue, at least acknowledging this pattern would be informative for the reader, as there are many results described and it can get lost in the mix; and this order of conditions, as far as I understand, is the one predicted by belief attribution.

11) In the first paragraph of the discussion the questions and answers the second two are reversed (the third answer replies Q2 and vice versa)

12) P. 38 lines 16 and 24 both start with "all in all"

13) The discussion from the appendix should be moved to the main text. If word limit prevents, some of the repetitive parts of the discussion can be streamlined (several of the paragraphs discuss the FB vs. IG / the seeing = knowing rule; factive mindreading, etc.

14) To my reply to point 15, the authors mention that in some measures FB and IG differed; to argue against the possibility that the test question biased children. However, this argument supposes that we take this finding seriously; which would mean that the findings overall are not consistent with the alternative rule. This is an inconsistency to be addressed – either much dampen the emphasis on the alternative rule, or explicitly acknowledge that this way if phrasing

the question may in fact have elicited interpretations based on this rule (since the prompt is pointing out the fact that the bear hasn't seen what Tom did)

15) Cf. my previous point 18, I still don't understand why, if children use a simpler mechanism that explains young children and infants' abilities such as seeing = knowing, there is an age effect in the implicit measures. Such rules are typically meant to explain data of infants as young as 6 months of age; why, then, is there an increase between 3 to 4 years?

===PREPARING YOUR MANUSCRIPT===

If you have been asked to revise the written English in your submission as a condition of publication, you must do so, and you are expected to provide evidence that you have received language editing support. The journal would prefer that you use a professional language editing service and provide a certificate of editing, but a signed letter from a colleague who is a fluent speaker of English is acceptable. Note the journal has arranged a number of discounts for authors using professional language editing services (<https://royalsociety.org/journals/authors/benefits/language-editing/>).

===PREPARING YOUR REVISION IN SCHOLARONE===

<https://royalsociety.org/journals/authors/author-guidelines/#supplementary-material> to include a suitable title and informative caption. An example of appropriate titling and captioning may be found at https://figshare.com/articles/Table_S2_from_Is_there_a_trade-off_between_peak_performance_and_performance_breadth_across_temperatures_for_aerobic_scorpions_in_teleost_fishes_/3843624.

Author's Response to Decision Letter for (RSOS-211278.R1)

See Appendix B.

Decision letter (RSOS-211278.R2)

Dear Dr Barone,

On behalf of the Editors, we are pleased to inform you that your Manuscript RSOS-211278.R2 "Do Young Children Track Other's Beliefs, or Merely their Perceptual Access? An Interactive, Anticipatory Measure of Early Theory of Mind" has been accepted for publication in Royal Society Open Science subject to minor revision in accordance with the referees' reports. Please find the referees' comments along with any feedback from the Editors below my signature.

Please submit your revised manuscript and required files (see below) no later than 7 days from today's (ie 11-Aug-2022) date. Note: the ScholarOne system will 'lock' if submission of the revision is attempted 7 or more days after the deadline. If you do not think you will be able to meet this deadline please contact the editorial office immediately.

on behalf of Dr Carina de Klerk (Associate Editor) and Essi Viding (Subject Editor)
openscience@royalsociety.org

Associate Editor Comments to Author (Dr Carina de Klerk):
Dear Dr. Barone,

Thank you for resubmitting your revised manuscript. I just have a few minor remaining comments that would need to be addressed before the manuscript can be published.

In some places in the manuscript the research questions are 'too big' relative to what the study does. The research questions relate to the state of the field while the manuscript focuses on one study. It is true that the task used in his manuscript is particularly informative, but the current study's results are not necessarily going to generalise to all other tasks in the field. I do think it's great to raise these questions in the introduction section as the theoretical rationale for the study, but the way they are presented at the beginning of the discussion section and summary paragraph may give the wrong impression that this single study is going to be able to provide a conclusive answer.

Please clarify in the main manuscript how the ignorance condition was introduced. I.e. how did you make it clear to the child that the bear didn't witness the original placement and transfer. It seems important to describe this in the main manuscript because the interaction between the experimenter and bear is described as a possible explanation for the findings in the discussion

section (e.g. on page 19 it is stated that “Another potential concern regarding the present study is that the IG condition may have been confusing (...). On the other hand, the way E1 and the bear interact may be taken to suggest that E1, something like an omniscient narrator in the scene, expects the bear to know where the object is”).

Relatedly, in their reply to the reviewer’s comments on the previous version of their manuscript the authors mentioned that you discuss how future studies may implement more refined versions of the ignorance conditions. It would be good if the authors could do this in a more explicit manner, i.e. acknowledge why exactly the IG condition may have been confusing and what could be done to remedy this in the future.

In the discussion section the authors suggest that the pattern of results in the ignorance condition may have resulted from some form of ‘pull of the real’ and that in cases of uncertainty it may be safer to go for the objectively good than for the objectively bad answer. Could a reality bias in some of the children also have influenced performance in the FB condition?

Like reviewer 2 requested please indicate how many of the effects were significant rather than describing results as ‘largely non-significant’ or ‘largely showing no differences’.

Please avoid using abbreviations unless they are very commonly used (like FB or TOM). So please just refer to anticipatory looking instead of AL (if possible within the word count).

Note that the acknowledgements section refers to ‘assistants’ to CogSci and BCCCD which presumably is meant to be ‘attendants’.

===PREPARING YOUR MANUSCRIPT===

one version should clearly identify all the changes that have been made (for instance, in coloured highlight, in bold text, or tracked changes);
a ‘clean’ version of the new manuscript that incorporates the changes made, but does not highlight them. This version will be used for typesetting.

Please ensure that you include an acknowledgements’ section before your reference list/bibliography. This should acknowledge anyone who assisted with your work, but does not qualify as an author per the guidelines at <https://royalsociety.org/journals/ethics-policies/openness/>.

If you have been asked to revise the written English in your submission as a condition of publication, you must do so, and you are expected to provide evidence that you have received language editing support. The journal would prefer that you use a professional language editing

service and provide a certificate of editing, but a signed letter from a colleague who is a proficient user of English is acceptable. Note the journal has arranged a number of discounts for authors using professional language editing services (<https://royalsociety.org/journals/authors/benefits/language-editing/>).

===PREPARING YOUR REVISION IN SCHOLARONE===

-- If you are requesting an article processing charge waiver, you must select the relevant waiver option (if requesting a discretionary waiver, the form should have been uploaded, see 'File upload' above).

-- If you have uploaded any electronic supplementary (ESM) files, please ensure you follow the guidance at <https://royalsociety.org/journals/authors/author-guidelines/#supplementary-material> to include a suitable title and informative caption. An example of appropriate titling and captioning may be found at https://figshare.com/articles/Table_S2_from_Is_there_a_trade-off_between_peak_performance_and_performance_breadth_across_temperatures_for_aerobic_scope_in_teleost_fishes_/3843624.

Author's Response to Decision Letter for (RSOS-211278.R2)

See Appendix C.

Decision letter (RSOS-211278.R3)

Dear Dr Barone

On behalf of the Editors, we are pleased to inform you that your Manuscript RSOS-211278.R3 "Do Young Children Track Other's Beliefs, or Merely their Perceptual Access? An Interactive, Anticipatory Measure of Early Theory of Mind" has been accepted for publication in Royal Society Open Science subject to minor revision. Please find the feedback from the Editors below my signature.

We invite you to respond to the comments and revise your manuscript. Below the Editors' comments we provide additional requirements. Final acceptance of your manuscript is dependent on these requirements being met. We provide guidance below to help you prepare your revision.

Please submit your revised manuscript and required files (see below) no later than 7 days from today's (ie 07-Sep-2022) date. Note: the ScholarOne system will 'lock' if submission of the revision is attempted 7 or more days after the deadline. If you do not think you will be able to meet this deadline please contact the editorial office immediately.

Kind regards,
Royal Society Open Science Editorial Office

on behalf of Dr Carina de Klerk (Associate Editor) and Essi Viding (Subject Editor)
openscience@royalsociety.org

Associate Editor Comments to Author (Dr Carina de Klerk):

Associate Editor

Comments to the Author:

Please edit the first sentence of the abstract. I understand that this change was made in response to my comment about the scope of the research questions addressed by this work, but I am not sure it makes sense to suggest that the study aims to contribute to clarifying three questions. The questions aren't being clarified by the work right? Instead the work aims to bring the field one step closer to answering these questions. Possible alternatives: "This paper aimed to contribute to answering three questions:" or "This paper was inspired by three questions currently occupying the field of infant theory of mind understanding". Please also update the final section of the introduction section based on my previous comment relating to the scope of the paper/research questions. The current work is still presented as if it can provide a conclusive answer to these big questions. While it is an informative study it is not going to 'solve' or answer these questions conclusively.

===PREPARING YOUR MANUSCRIPT===

one version should clearly identify all the changes that have been made (for instance, in coloured highlight, in bold text, or tracked changes);

If you have been asked to revise the written English in your submission as a condition of publication, you must do so, and you are expected to provide evidence that you have received language editing support. The journal would prefer that you use a professional language editing service and provide a certificate of editing, but a signed letter from a colleague who is a proficient user of English is acceptable. Note the journal has arranged a number of discounts for authors

using professional language editing services
(<https://royalsociety.org/journals/authors/benefits/language-editing/>).

===PREPARING YOUR REVISION IN SCHOLARONE===

-- If you are requesting an article processing charge waiver, you must select the relevant waiver option (if requesting a discretionary waiver, the form should have been uploaded, see 'File upload' above).

-- If you have uploaded any electronic supplementary (ESM) files, please ensure you follow the guidance at <https://royalsociety.org/journals/authors/author-guidelines/#supplementary-material> to include a suitable title and informative caption. An example of appropriate titling and

captioning may be found at https://figshare.com/articles/Table_S2_from_Is_there_a_trade-off_between_peak_performance_and_performance_breadth_across_temperatures_for_aerobic_sc_ope_in_teleost_fishes_/3843624.

Author's Response to Decision Letter for (RSOS-211278.R3)

See Appendix D.

Decision letter (RSOS-211278.R4)

Dear Dr Barone:

I am pleased to inform you that your manuscript entitled "Do Young Children Track Other's Beliefs, or Merely their Perceptual Access? An Interactive, Anticipatory Measure of Early Theory of Mind" is now accepted for publication in Royal Society Open Science.

Please remember to make any data sets or code libraries 'live' prior to publication, and update any links as needed when you receive a proof to check - for instance, from a private 'for review' URL to a publicly accessible 'for publication' URL. It is also good practice to add data sets, code and other digital materials to your reference list.

Royal Society Open Science is a fully open access journal. A payment may be due before your article is published. Our partner Copyright Clearance Center's RightsLink for Scientific Communications will contact the corresponding author about your open access options from the email domain @copyright.com (if you have any queries regarding fees, please see <https://royalsocietypublishing.org/rsos/charges> or contact authorfees@royalsociety.org).

on behalf of Dr Carina de Klerk (Associate Editor) and Dr Essi Viding (Subject Editor).

<https://www.facebook.com/RoyalSocietyPublishing/>

Appendix A

Dear Dr. de Klerk,

Thank you very much for your letter regarding our manuscript. We are very happy to hear that the reviewers and you thought the paper addresses an important and timely topic. And we are very grateful for your and the reviewers' comments and suggestions which we found very helpful for improving the manuscript. Please find attached a thoroughly revised and improved version in which we carefully address the central issues raised by the reviewers. Below, we detail how we responded to each issue (in red).

We are looking forward to hearing back from you.

With best wishes,

Pamela Barone

Associate Editor Comments to Author (Dr Carina de Klerk):

Comments to the Author:

As you will see from the enclosed comments both reviewers find the work interesting and believe that it adds to the ongoing debate about early theory of mind development. However, both reviewers also raised a number of important issues that need to be addressed. In particular, both reviewers commented on the way the results are described, suggesting you should refrain from using terms like 'marginal' to describe non-significant findings and both suggest you could use Bayesian analyses to clarify null results. Furthermore, reviewer 2 notes that a more balanced representation of the literature should be presented in the introduction and discussion section. This reviewer additionally points out several important issues with the interpretation of the findings, in particular relating to the ignorance condition and the strength of the evidence for the seeing=knowing rule, which will need to be addressed in a revision.

Reviewer comments to Author:

Reviewer: 1

Comments to the Author(s)

I have read this manuscript with great interest, as it nicely adds to the ongoing debate about the existence and nature of early, implicit mentalizing abilities. The comments I had mainly concern the reporting of the results, as I will outline below.

Quite a few results are described as ‘tending to be significant’ / ‘marginal effects’ / ‘marginally differing’, etc. Such terms can be misleading, so it is best to simply report such results as non-significant. A Bayesian analysis could be computed if the authors wanted to comment on the strength of evidence for either null or alternate model. In addition, the authors often perform follow-up pairwise comparisons even in the absence of significant main effects (and it appears these are not corrected for multiple comparisons). This is not exactly common practice, so unless these comparisons were planned, I would avoid it as much as possible in order not to over-interpret the results.

Response: Thank you for these suggestions which we now happily follow: In the revised version, we do report results only as “significant” or “not significant”, and do not report follow-up comparisons on non-significant effects.

I was wondering if the authors have any data on the children’s responses to the experimenter’s question about the bear’s perceptual access. This seems a crucial part of the design: whether or not the bear saw what happened, is made explicit to the child. Was there any relation between their initial answer to this question and their performance?

Response: Thank you for raising this crucial point. In response, we now clarify the following: We here followed the rationale and method of Rubio-Fernandez (2013; “These prompts were intended to help the child keep track of the [agent]’s perspective”). So, given served more as a prompt than as a proper question, we actually did not formally code children’s responses.

Another question I had was why ‘gender’ was included as a factor in the different models, as this is not really justified. Did the authors have any expectations of gender differences?

Response: Thank you for raising this point. No, we did not have any expectations about that. We just wanted to control as many variables as we could.

Minor comments:

- There are some spelling / language issues, e.g.:

Summary: ‘How robust and reliable are’ -> ‘How robust and reliable is’;

Summary: ‘behaviors towards the agent is measured’ -> ‘behaviors towards the agent are measured’;

Methods (3.3): ‘look one side over the other’ -> ‘look at one side over the other’;

Appendix A (3): ‘made of cupboard’ -> ‘made of cartboard’?

Please check for such mistakes throughout.

Response: Thanks for noting the previous mistakes. We have corrected them.

- Methods, 3.1: 'Around their 4th birthday is still relatively young, not all children may have developed explicit ToM abilities fully. Can the authors comment on how this may have influenced results?

Response: The older sample had 31 children around their 4th birthday (age range = 45 – 51 months; mean age = 47.78 months). As we can see from the Results (4.4) the older group passed the explicit FBT above chance level (81% correct).

- Results, 4.1.1: The description of the order effects was not quite clear to me. Did they look more at the full box on all three conditions for 'IG-TB-FB' order, or on one/two conditions in particular?

Response: We are afraid that this may rest on a misunderstanding. As we stated in the model, there was no effect of condition order in anticipatory looking = $F(5, 72) = .584, p = .712, \eta^2 = .007$. In the last paragraph, we reported separate tests that indicate that participants preferred to look at the full box in the TB condition, but they showed no preference in the FB or IG conditions.

- Results, 4.5 has the same title as 4.4 – this might have to be rephrased?

Response: Thank you for spotting what is indeed a mistake. In response, we have changed the subtitle in 4.5 to "Correlations among measures".

- Figure 2: the stripes indicating which conditions significantly differ from one another seem to be misplaced.

Response: Thanks for indicating that issue. We realize now that the figure was changed in the submission platform modifying the stripes as a result. We will attach the file separately in this new submission, so the stripes should not move.

- Appendix A, '2. Participants' -> this seems to be the wrong title.

Response: This is, in fact, wrong. It has now been changed to "Procedure".

- Appendix B (2): Were the differences between the different filler trials significant? (Worse performance after IG condition?)

Response: Thanks for raising this question. There were no significant differences between the different filler trials (Cochran's Q test, $\chi^2(2) = 2.235, p > .05$). We have now clarified this issue in Appendix B, together with the other analyses of filler trials.

Reviewer: 2

Comments to the Author(s)

The present study is a conceptual replication of a combined interactive-spontaneous and verbal- theory of mind task, based on Garnham and Perner (2001). The goals of the study are ambitious: probe the replicability of the paradigm, extend it, and address the theoretical debates surrounding the interpretation of children's behavior in the task. This is an important endeavor, and in general I very much support replication projects to be published. In the present form, however, I find that a more balanced picture of the literature as well as a more detailed and nuanced discussion of the current findings should be presented. Below I list some points to address.

Introduction:

1. The original study is not presented in the introduction, and thus is not clear in what sense this is a replication: what paradigm and setup did the original include, and what exactly did they analyze and find.

2. This should/could be done instead of the rather general introduction of various implicit ToM paradigms, e.g. VoE is not strictly relevant for the current paper, whereas e.g. not much is mentioned about the interactive tasks. (in addition, Kiraly et al, PNAS, is mentioned as a non-replication but if I remember well they replicated the original findings).

Response to 1. & 2.: Thank you for raising these important points. We agree, and in response, we have now revised the introduction in the following ways: we have shortened the review of VoE findings, and instead now have included a new paragraph with the most important information about the original Garnham & Perner study.

3. Relatedly, some of the literature would be very relevant to introduce, on tasks that are similar to the Garnham and Perner one, e.g. studies by Scott et al, other interactive tasks, or paradigms that for example have spontaneous measures to verbal stories.

Response: Thank you for referring us to these studies which we now also cite in the Introduction.

4. There is also a large literature on the verbal ToM tasks and their critique, stability of passing within an individual child, papers that disagree with the fact that verbal FB tasks should measure metarepresentations or that it should be used as a litmus test. There are also studies showing that the capacities involved in these tasks may develop further after

age 4. These should be at least mentioned, especially since the current study does not probe metarepresentations directly, and since debates around the implicit measures are discussed in some detail. It is not the case that the verbal or “standard” FB tasks stand without critique and I would be helpful not gloss over these arguments, especially when looking at the relation of this task to other measures, where it may be relevant to think about what capacities may be involved in the different tasks. If not in the introduction, then in the discussion considering these would be informative.

Response: We are grateful that you bring up this important issue. Unfortunately, discussing these crucial topics in detail would be well beyond the scope of the present paper. But in the revised version we now at least mention some of these concerns and cite relevant work.

5. Regarding the theoretical claims in the introduction: if the authors aim to include debates on metarepresentations, it would be relevant to explain which implicit tasks are proposed to measure these and why, as currently it is simply stated (cf. p. 3 line 36-40).

Response: We are afraid that this rests on some misunderstanding. Our aim here was not to enter into foundational debates regarding meta-representation, but to review which tasks are generally considered to strictly require it and which don't, and how these fare with respect to replicability. We clarify this point now in the revised version.

6. In order to situate it in the (non)-replication literature and findings, it could be relevant to mention, why did the authors choose this particular paradigm, what does it add to previous ones, and how does it differ from the other (non-) replications?

Response: Thank you for pressing us to justify more clearly the choice of task. Upon re-reading the paper, we agree that we did not make this point clear enough in the previous version. Therefore, we now clarify the underlying rationale: we chose this particular paradigm because it allows us to do several crucial things at once: to combine different implicit measures (looking and interactive behaviours), to add a crucial ignorance condition, and to compare children's performance to a standard FB task – all of which together make this task particularly informative.

Results:

7. In general, in order to probe these various different effects, this study may be underpowered. This is also suggested by the many non-significant “trend” effects (which are generally advised not to be used as a term, as in frequentist statistics it is somewhat unclear what it means), which may come from the lack of power. Bayesian statistics could also help (though not solve) the absence of effects, where instead the confidence in the null could be

expressed. I am aware of the power calculations the authors did before the study, but that did not include the ignorance condition or the hesitance measures. In fact, it seems to me that those effects that were based on the power calculations largely replicated, but I may be mistaken.

Response: Thank you for raising this point. In response, we now clarify: The Ignorance condition and the hesitance measures were not included in the power calculations since they were not done in the original study which served as the basis for the power analysis.

8. For the first look (and perhaps for the other measures too) it would be useful to do a first trial analysis. This is also shown by the interaction between order and first look, which is hard to interpret, and the authors do not return to. If there is not enough power in the current sample, then perhaps in a follow-up to strengthen and clarify these results.

Response: Thanks for mentioning it. We have now included a first trial analysis of all the measures in the appendix.

9. In the first look, it is mentioned that there is a significant difference between FB and IG, which if true, then the claim in the discussion (p. 9, line 9) is not correct and there are differences between FB and IG. But it seems to me this may be a typo?

Response: Thank you very much for pointing this out. We now clarify in response: It is true that the only difference between FB and IG was in first look, and we adapted the corresponding parts in the discussion accordingly.

10. In DLS, $p = .087$ is referred to as tending to significance but $p = .11$ is not (former is the difference between TB and IG, latter between FB and IG), I am sympathetic to not only considering effects below the $p = .05$ cutoff, but then some other way of discussing/analyzing the results should be added (e.g. Bayesian statistics).

Response: Thank you for pointing out this mismatch. In the revised version, also following the suggestions of Reviewer 1, we do report results only as “significant” or “not significant”.

11. In catching behavior, children in IG chose the full box above chance, not consistently with the seeing=knowing rule that is proposed.

Response: Thank you for raising this point. Yes, it is true that children did choose the full box more often than the empty box in IG – but for present purposes, the crucial result pertains to the comparison between FB and IG conditions. Proper belief ascription would predict systematic differences between the conditions, but none such were found.

12. While in the latency, catch-shifts, and look-shifts measures the effects are generally not significant, all three follow the TB < FB < IG order. Perhaps the two shifts could be merged into a composite measure, but in any case, this is a pattern that may be worth commenting on. I am not convinced that this shows that there was equal amount of hesitance in FB and IG. In addition, neither of these 3 hesitance measures show that TB and FB differ from each other, so should we then conclude that children equated TB and FB?

Response: Thank you for this suggestion. In response, we clarify the following: First of all, the order in catch and look shifts over conditions are somewhat different so that they cannot be combined or merged (Catch-shifts follow the IG < TB < FB order (more shifts in IG, although no difference among conditions). Look-shifts follow the FB < IG < TB (more shifts in FB)). Second, we now are more careful in not drawing any conclusions regarding evidence for absence from absence of evidence regarding condition differences (or absence thereof).

13. Passers in first look appear to me to behave consistently with belief ascription and not seeing = knowing: above-chance to full box in TB, above-chance to empty in FB, and chance in IG. Non-passers on the other hand look at chance in FB. This pattern should be commented on. (similarly in DLS, when considering younger vs. older kids instead of passing on explicit task, in the older group the entire pattern seems to me consistent with belief attribution). Relatedly, the authors report that first look in the FB condition was associated with the explicit FB performance. Of the things that could be related between the two tasks, this is arguably one the most relevant to probe whether the interactive and explicit measures are related.

Response: Thank you very much for raising this very important point. We fully agree that this is a crucial piece of information that was not sufficiently clearly explained or emphasized in the previous version. In response, we now explain and discuss much more clearly and explicitly: The key finding (regarding first looks, standard FB passers' first look is above-chance to full box in TB, above-chance to empty in FB, and chance in IG; while non-passers' first look is at chance both in FB and IG) really does suggest that successful performance across the three conditions of the present task reflects the same competence as tapped in standard explicit tasks.

Discussion:

14. Overall, it seems to me that the ignorance condition may not have worked as intended. It may stem from the implementation: the interaction between E1 and the bear may indicate

that the bear will know where she puts it, and then when Tom relocates it, it effectively turns into a false belief. But in fact, in some of the measures, children act as if the bear had a true belief (e.g., in catching behavior, and especially the younger children). This potentially confusing nature of the IG condition may be reflected in the hesitance measures, where on all three the IG condition shows most hesitance. This seems especially true in non-passers.

Response: Thank you for bringing up this critical point. In response, we now discuss how future studies may implement refined versions of ignorance conditions.

15. Is it possible that the question: “The bear hasn’t seen what Tom did there, has he?” prompted children to use a seeing-knowing strategy? This would also make the FB and IG conditions effectively similar to each other.

Response: Thank you for raising this point. While it is not impossible, we think this is not very plausible for the following reasons: It is true that the prompt is the same for FB and IG conditions. However, the results from our main analyses (taking the whole group together) do not suggest that they were treated in the same way: in first look, there’s a significant difference between FB and IG and children caught the bear at the end of the full slide significantly above chance in IG, while they did it at chance in FB. Then, there are some measures that might indicate that children did not treat FB and IG similarly.

16. On p. 8 line 55-56 it is stated that in first look children showed a tendency to look first to the full box in IG, but from what I gather from the results, this was not the case.

Response: Thank you very much for pointing this out. We have now clarified, also following your suggestion number 9, that in first look there is a difference between conditions (FB and IG) but if we analyse them separately, children’s first look was at chance in both conditions.

17. Throughout, it is suggested that the verbal/standard FB task is harder than the interactive ones, because the latter involves implicit ToM which is simpler/easier. But in fact we cannot be sure that the interactive part involves “implicit” processing especially at this age; and if we consider that some (or all) of these children may undergo some cognitive changes related to the verbal ToM (whatever they may be, unrelated to exact theoretical commitment) then they may be reasoning explicitly and then have to plan their actions based on that, which may make the interactive part more difficult. A more elaborate discussion on the measures could benefit the paper.

Response: Thank you very much for this suggestion. We fully agree that these are issues that deserve to be addressed. Doing so here in detail, however, would go well beyond the scope

of the present paper. But we have some included some discussion of these points in the General Discussion.

18. I am a little confused about the conclusions. The interactive measures are robust and increase with age, but they measure a simpler mechanism - so this simpler mechanism increases with age similarly to the metarepresentational capacities? It may be informative to relate this to other literature on knowledge-ignorance or seeing=knowing.

Response: Thank you for this critical point. It made us realize, indeed, that we did not make the crucial points sufficiently clear in the previous version. We therefore clarify the following in the revised version: Children can respond to this task (as to many ToM tasks) by using different types of reasoning, including a simpler (factive, relational etc.) and fully meta-representational one. Theoretically, this is in line with many accounts including conceptual change, dual process or “factive Theory of Mind” accounts. Empirically, it is in line with other recent studies that suggest that older children show qualitatively different patterns of responses compared to younger ones (Grosse Wiesmann et al., 2017; Wenzel et al., 2020).

19. Relatedly, how does one explain those who passed the explicit FB and didn't pass the implicit ones: they have metarepresentations but cannot apply these when they need to catch someone?

Response: Thank you for raising this point. In response, the crucial point for current purposes is the following: what matters is whether the two tasks are systematically related (and it turns out that there are such relations for some implicit measures (first look), but not others). But of course, such correlations, if there are any, are not perfect in light of noise (and thus, it is to be expected that some children show such patterns as the one mentioned here in this reviewer comment).

20. The 'seeing leads to knowing mechanism could be elaborated on. For example, why doesn't not seeing lead to being mistaken? How do children represent others' ignorance?

Response: Thank you for bringing up this important question. In response, the following should be clarified: There are several ways in which seeing=knowing rules have been proposed and spelled out. One according to which non seeing indeed leads to mistakes (Fabricius et al., 2010; 2021), and another one which does not make such additional assumptions (Phillips et al. 2021). It is the latter that is relevant for current purposes. We make this now clearer in the Introduction and Discussion.

21. On p. 8, line 57 it is mentioned that the various measures in the interactive task were related to each other. I apologize if I missed this, but I did not find the analyses this is based on.

Response: Many thanks for noting this error. In fact, they were not included in the previous draft, but the revised version has been modified accordingly. On the one hand, we calculated correlations among the various measures in the interactive FB task and all of them were statistically significant. On the other hand, we tested the correlation between each of the implicit measures and children's performance in the standard FB task. Only children's first look in the FB condition was associated with the standard FB task (Cramer's $V=.25$, $p=.025$). All other correlations were not significant.

Overall, the results seem mixed and while some results replicate, others don't (e.g., I think in the original study the looks were significantly above chance in the belief-direction also in FB); and these should be detailed and discussed. Additionally, in order to make larger theoretical claims, a more fine-grained and critical discussion of the results should be included, with those patterns also acknowledged that may warrant more nuanced interpretations.

Currently, it does not seem to me that it is entirely clear what the data show related to the three targeted questions. To answer question (1) a description of the original findings would be needed, and discussing the current ones in more detail; regarding (2) it appears that the manipulation may not have worked ideally, yet in some measures there may be some differences between FB and IG, whereas in contrast in others the pattern of response appears not consistent with a seeing=knowing rule, and for point (3) I could not find where the relation between interactive measures are reported (again, apologies if I missed it), and in some measures there is a relation with the explicit task, or response pattern consistent with belief attribution in the implicit tasks, specifically in explicit-passers only. Together, addressing these would contribute greatly to the existing literature and move towards clarifying replicability of ToM paradigms and cognitive foundations of ToM.

Smaller remarks

22. Figure 2, especially Fig 2D, appears somewhat scrambled and the error bars (?) are shifted. Or if that's not the case then I am not sure how to interpret the horizontal lines across the figures.

Response: Thanks, there was a problem in the submission of the figures, and they have been changed in the merged submitted version. This problem will be solved when submitting the images separately.

23. In supplemental Figure 1 the labels for younger children appear somewhat scrambled.

Response: The same response as in the previous comment applies here.

24. Sections 4.4 and 4.5 have the same name (“Standard FB task”)

Response: Thanks, we have modified the heading 4.5.

Appendix B

Dear Dr. de Klerk,

Thank you very much for your letter regarding our manuscript. We are very happy to hear that the reviewers think that our paper has improved. And we are very grateful for your opportunity to address the remaining issues that the reviewers have identified in one final round of revision. Please find attached a thoroughly revised and improved version in which we carefully address the central issues raised by the reviewers. Below, we detail how we responded to each issue (in red).

We are looking forward to hearing back from you.

With best wishes,

Pamela Barone

Associate Editor Comments to Author (Dr Carina de Klerk):

Associate Editor: 1

Comments to the Author:

Dear authors,

The revised manuscript was sent back to one of the original reviewers and although they indicate that the manuscript is much improved, they also identified several remaining issues that would need to be addressed before the manuscript is suitable for publication. Although Royal Society Open Science does not normally allow for multiple rounds of revision, the Subject Editor and I believe it should be possible to address the remaining issues in one final round of revision and we therefore invite you to revise and resubmit your manuscript. Please make sure you address all the reviewer's remaining comments in your revision.

Reviewer comments to Author:

Reviewer: 2

Comments to the Author(s)

Generally, the manuscript has improved substantially, is more clear, and I applaud the changes the authors made, and appreciate most of their responses. I have one broad, major issue still, which is that I find the ignorance condition, as I pointed out at various points, not convincing. It could be partially the implementation, and largely sample size, and potentially other factors. If there was a way to show that this sample

size was big enough to find these effects then I this argument would not hold as much; but currently there is no such evidence provided, and many (if not most) of the reported effects are in the right direction just not significant. I believe that while the IG condition was a creative addition that could yield strong results, we cannot be sure if it succeeded. As such, I find the point with the replication fully convincing, the relation with the explicit task moderately (as 1 out of 3 measures show a relation); and the alternative theoretical explanation not convincing, and suggest to revise the abstract and discussion accordingly, and provide a balanced presentation of all results (e.g. if 1 of 3 effects is significant then instead of “largely not significant” – simply the number can be described).

Response: We are glad that you liked the revision of our manuscript. We also like to thank you again for stressing your concerns about the power in our study concerning the comparison of the IG to FB and TB conditions. We completely agree with you about being cautious with interpreting null results, even more given that the sample size calculation was based on the original Garnham & Perner (2001) study that did not include the IG condition. We would like to respond in two ways here.

First, as we report in more detail in Appendix A, our final sample was based on the power analysis of the smallest effect of the original study ($OR=4.3$ for the comparison between interactive behaviour and explicit answer). However, the original effects varied a lot depending on contrast (note, e.g., that they did not directly compare TB and FB scenarios, but mostly interactive and explicit answers) and type of measure. For example, for the first look the original effect size was much bigger, leading to a projected sample size of 44. Thus for the case of first looks our sample size is almost double compared to the projected one, which makes it less likely that the IG/FB contrast was underpowered here.

Second, even though we agree that on the basis of belief attribution one would predict a smaller effect for the IG/FB and IG/TB comparisons than for the FB/TB comparison for first look, DLS and catching behaviour measures, the situation is quite different for the measures of uncertainty. If the child attributes a belief to the agent, whether true (TB) or false (FB), she should be certain in her response (i.e., that the agent will come down the slide of the full or empty box, respectively). There is no reason to assume that she is less certain in the FB case than if the agent has a TB (see also Ruffman et al., 2001). However, the child should hesitate in her response in the IG condition in which there is no determinate expectation and no correct response and the child can thus only guess. Therefore, for the uncertainty measures we would not expect a difference between TB and FB, and a strong effect for the TB/IG and FB/IG

comparisons. Accordingly, the potential power issue that you mention only arises for the first look, DLS and catching behaviour measures, but does not apply for the measures of uncertainty (see also our response to point 10).

We completely agree that our argumentation would be stronger if we could substantiate it with number. However, we refrained from running post-hoc power analyses as there are several issues with these (see e.g., Lakens, 2014; Zhang et al., 2019). The best practice from our point of view would be a Bayes null hypothesis testing approach, however, this would need to go along with a much bigger sample size. We now added this discussion as a limitation explicitly in the manuscript.

I also have several smaller points which I outline below (some are related to my above point).

1) Now I understand better the rationale of the present study, but the abstract and the intro do not reflect this, except the last sentences of the introduction. While replication is generally a reasonable endeavor, there are also many tasks out there by now. I strongly believe that it is necessary to justify and define from the get-go what the scope and goal here is.

Response: Thank you for pushing us in making the rationale clearer from the very beginning of the article, which we now do in the revised version.

2) My point with raising the power analysis (point 7 in my previous review) is not simply to add a comment in the methods that IG condition was not added to the power analysis. I think it is important to mention as limitation and dampen conclusions throughout which refer to effects that may be underpowered. The authors should add this in the main text and add a section discussing that some of the contrasts may have depended on this. As I mentioned, to me it looks that those effects that were sufficiently powered based on the power calculation, did replicate, in line with the possibility that not sufficient power may be an issue in the others. Perhaps post-hoc power analyses could be performed for the critical comparisons.

Response: Thank you again for raising this concern, which we replied to in the general response (see above).

3) In point 11 in my previous review I pointed out that children in the IG chose the full box above chance which is not consistent with the proposed rule. While it may be that the crucial contrast for the authors is FB vs. IG, but if they wish to make theoretical claims regarding what cognitive mechanisms may underlie children's behavior then these should be consistent with the observed pattern.

Response: Thank you for raising this point again. It made us realize that we should have been clearer in describing the different mechanisms that might underlie children's behaviour, especially in the IG condition. We agree that following the "seeing=knowing" rule, children should be at chance in the IG condition for the first look, DLS, and catch measure, as -given the agent's ignorance- both sides are equally likely. However, what we actually find is that for the catch measure, children chose the side of the full box more frequently than the side of the empty box.

Now, does that speak against the "seeing=knowing" rule? We don't think so. We believe what we see here is a mixture of different processes, resulting from the fact that we put children in a difficult position: we asked them to make a choice even though there was no correct answer. One way to go in such cases is the real location (also been termed the "pull of the real" (Carpenter et al., 2002) or "reality bias" (Southgate et al., 2007): even though the agent doesn't know about it, there is an objectively good reason to choose the full box: it actually contains the reward (see e.g., Perner & Roessler, 2012).

We now added this important point in the discussion.

4) Why did the authors delete the Wilcoxon tests? (I am not criticizing, merely wondering)

Response: Thanks for asking. This was done in reply to a comment raised by Rev 1 to not perform follow-up pairwise comparisons in the absence of significant main effects, which is the case for uncertainty measures.

5) I agree with Rev 1 that gender is unnecessary to add, unless there is a hypothesis for it.

Response: Thank you for raising this point. We completely agree with you that we should not have added gender in our model since we did not have an expectation about it. We have now adapted all our models accordingly removing gender and reporting the results in a simpler and cleaner way by analysing whether there are

differences between conditions regarding all our dependent variables: first look, DLS, interactive behaviour, and uncertainty measures. The rationale for doing that is that we can remove all the parameters that may obscure the interpretation of what is actually happening in each condition, and, in the additional analyses, run the same models for a) first trials, b) age groups, and c) standard FB task passers vs non-passers.

The overall picture remains the same but there are two changes regarding FB-IG comparisons. 1) When the whole group is considered, children were as likely to look first at the full box in both FB and IG conditions. 2) Regarding interactive behaviour, children were more likely to choose the full box in the IG condition than in the FB condition. In the previous model with all the predictors included, it was the opposite (no difference in catch but more likely to look first at the full in the IG). Looking at the data, the new results nicely resemble what is seen in the graphs. We thus believe that the change was due to the exclusion of the parameters of the previous models (age, or performance in the standard FB task) that we can now analyze in the additional analyses. At the same time, we can evaluate in a cleaner way the type of responses given in each condition by the whole group.

6) I understand now that the results on the significant correlations within interactive measures were not included, and now understand what the discussion referred to. However, since this has been also subject to much debate, I would find it important to comment on the fact that within the implicit measures the correlations were significant; definitely in the discussion, but perhaps even in the abstract; especially considering that the authors mention this as one of the goals of the present paper / using this task

Response: Thank you for these suggestions which we now happily follow: In the revised version, we added a statement about this in the abstract.

7) The authors not find the effect by Rakoczy et al re: FB-passers start failing TB – is it perhaps relevant to comment on?

Response: Thank you for raising this interesting point. The TB effect is defined as drop in performance in the TB control condition of the standard change of location task once children pass the FB condition (see e.g., Oktay-Gür et al., 2018). Currently, the most plausible explanation for this paradoxical finding is that it reflects pragmatic confusion on the part of the child. The TB tasks poses a trivial academic test question

about a rational agent's perspective that is shared by all people involved ("Why is the experimenter asking me this dumb question, even though we all share the same knowledge? I guess there must be something more to it."). In line with this explanation several studies to date could show that the effect disappears once the scenario and the test situation are made less trivial (see Oktay-Gür et al., 2018; Rakoczy & Oktay-Gür, 2020; Schidelko et al., 2022). In particular, the effect goes away, and children have no difficulty with TB tasks whatsoever once the task is presented in non-verbal, interactive format in which there is no test question that could engender pragmatic confusion (Rakoczy & Oktay-Gür, 2020; Study 1). Much in line with this argumentation, the present study involved tasks in largely non-verbal, interactive format without an explicit test question that would yield pragmatic confusion. The present findings thus nicely fit with, and corroborate, previous results about the pragmatic interpretation of the strange TB performance patterns.

However, we refrained from discussing these implications in the manuscript since they clearly would require quite some extensive backgrounding (on pragmatics etc.) and would go well beyond the scope of the present paper.

8) The discussion is more balanced now, but still not completely. E.g., on p.37, lines 48-49 the authors write that children's performance should differ drastically between FB and IG. This is not really the case, as in IG we'd expect chance and in FB an effect, but this may need sufficient power to reach as it is not an opposite direction of effect expected (which is in FB vs TB). (This would be a good place to mention the issue of power.) There is also a lot of the discussion addressed to the FB vs IG, arguing still, and with increasingly strong wording, for alternative mechanisms. This on one hand is repetitive, and on the other hand there are other aspects of the results to mention while this is toned down- currently it is worded more strongly than what i'd say the results warrant.

Response: Thanks for raising this point. In response, we now toned down the wording in the discussion concerning the expected effects for the FB/IG contrasts and (as mentioned above) added the discussion of a potential issue of power.

9) Currently in the appendix B, 5.1.1 first look It says FB vs. IG $p=0.018$, which would be significant; and their pattern is consistent with belief ascription not seeing = knowing rule; and not consistently with that the discussion in the appendix says.

Response: Thanks for noting this issue because we found a typo in this paragraph. As we are making multiple comparisons, post hoc analyses are conducted against a Bonferroni-adjusted alpha level of 0.017 (.05/3). Then, a p-value of 0.018 is not significant.

10) In my previous point 12, according to common notation, I used $TB < FB < IG$ to express there was more catch-shift in FB than in TB, and more in IG than in FB ($TB < FB$ refers to FB having more than TB). To me it still looks like it is $TB < FB < IG$ in all three, e.g. latency in TB is 453.17ms, in FB 603.37ms, and in IG 745.93; the catch-shift in $TB=.16$, $FB=.19$, and in $IG=.26$, suggesting more catch-shift in IG; and similarly in look-shifts $TB=.65$, $FB=.93$, and $IG=1$; again corresponding to the same order. If I am reading these wrong I would appreciate if the authors explained the misunderstanding by referring to the numbers. I understand that (some/all? of) these are not significant but as again this may be a power issue, at least acknowledging this pattern would be informative for the reader, as there are many results described and it can get lost in the mix; and this order of conditions, as far as I understand, is the one predicted by belief attribution.

Response: Thank you for restating this issue. We apologize for the previous misunderstanding. It is true that for all three uncertainty measures there is a descriptive trend in the form of $TB < FB < IG$. However, none of the comparisons between conditions reached significance, so that we think one should be careful in interpreting this pattern. Furthermore (as we also noted in our general response), and maybe even more importantly, we do not fully agree that this trend ($TB < FB < IG$) is the one predicted by belief attribution. If the child attributes a FB to the agent, she should be certain in her response (i.e. the agent will come down the slide of the empty box). We would thus not expect a difference in the uncertainty measures between TB and FB, but a strong effect compared to the IG condition, leading to a $TB = FB < IG$ pattern. We now realized that we did not make these predictions explicit in the manuscript, which we now updated accordingly.

11) In the first paragraph of the discussion the questions and answers the second two are reversed (the third answer replies Q2 and vice versa)

Response: Thanks. We have now modified the order of the answers.

12) P. 38 lines 16 and 24 both start with "all in all"

Response: Thanks for identifying this.

13) The discussion from the appendix should be moved to the main text. If word limit prevents, some of the repetitive parts of the discussion can be streamlined (several of the paragraphs discuss the FB vs. IG / the seeing = knowing rule; factive mindreading, etc.

Response: Thank you. Our logic was that the discussion in the Appendix covers the results presented in the Appendix. However, we now follow your suggestion and have included the main points in the main text.

14) To my reply to point 15, the authors mention that in some measures FB and IG differed; to argue against the possibility that the test question biased children. However, this argument supposes that we take this finding seriously; which would mean that the findings overall are not consistent with the alternative rule. This is an inconsistency to be addressed – either much dampen the emphasis on the alternative rule, or explicitly acknowledge that this way of phrasing the question may in fact have elicited interpretations based on this rule (since the prompt is pointing out the fact that the bear hasn't seen what Tom did).

Response: Thank you for raising this issue again. As we claimed before, we do not think that the way we phrased the question in the IG condition could have elicited this interpretation, as similar prompts have been used in other studies (see e.g., Rubio-Fernández & Geurts, 2013). Anyway, we have now discussed the possible explanation of such difference in performance that emerged in one FB-IG comparison (IG condition might have elicited a “pull of the real”, or such difference in performance is driven by the behaviour of the group of non-passers). In any case, we acknowledged it in the discussion and, accordingly, made our argument softer.

15) Cf. my previous point 18, I still don't understand why, if children use a simpler mechanism that explains young children and infants' abilities such as seeing = knowing, there is an age effect in the implicit measures. Such rules are typically meant to explain data of infants as young as 6 months of age; why, then, is there an increase between 3 to 4 years?

Response: Thank you for insisting on this critical enquiry. In response, we now try to make the crucial points clearer and more transparent. The basic insight is that a given type of (implicit) task does not necessarily tap a fixed given type of (implicit) process.

This has long been stressed, for example, in memory research, where assumptions that types of tasks (implicit vs. explicit) nearly map onto types of processes (implicit vs. explicit) have long been given up (e.g., Jacoby, 1991). Rather, some types of tasks (e.g., priming) may tap both implicit and explicit processes in different circumstances to different degrees. Similarly, in the current context, children can respond to the current interactive task (as to many ToM tasks) by using different types of reasoning, including a simpler (factive, relational, etc.) and fully meta-representational one. Theoretically, this is in line with many accounts including conceptual change, dual process or “factive Theory of Mind” accounts. Empirically, it is in line with other recent studies that suggest that older children show qualitatively different patterns of responses compared to younger ones (Grosse Wiesmann et al., 2017; Wenzel et al., 2020).

References

- Carpenter, M., Call, J., & Tomasello, M. (2002). A new false belief test for 36-month-olds. *British Journal of Developmental Psychology*, *20*(3), 393–420. <https://doi.org/10.1348/026151002320620316>
- Garnham, W. A., & Perner, J. (2001). Actions really do speak louder than words - But only implicitly: Young children’s understanding of false belief in action. *British Journal of Developmental Psychology*, *19*(3), 413–432. <https://doi.org/10.1348/026151001166182>
- Grosse Wiesmann, C., Friederici, A. D., Singer, T., & Steinbeis, N. (2017). Implicit and explicit false belief development in preschool children. *Developmental Science*, *20*(5), e12445. <https://doi.org/10.1111/desc.12445>
- Jacoby, L. L. (1991). A process dissociation framework: Separating automatic from intentional uses of memory. *Journal of Memory and Language*, *30*(5), 513–541. [https://doi.org/https://doi.org/10.1016/0749-596X\(91\)90025-F](https://doi.org/https://doi.org/10.1016/0749-596X(91)90025-F)
- Lakens, D. (2014). *Observed power, and what to do if your editor asks for post-hoc power analyses.* The 20% Statistician. <http://daniellakens.blogspot.com/2014/12/observed-power-and-what-to-do-if-your.html>
- Oktay-Gür, N., Schulz, A., & Rakoczy, H. (2018). Children exhibit different performance patterns in explicit and implicit theory of mind tasks. *Cognition*, *173*, 60–74. <https://doi.org/10.1016/j.cognition.2018.01.001>

- Perner, J., & Roessler, J. (2012). From infants' to children's appreciation of belief. *Trends in Cognitive Sciences*, 16(10), 519–525. <https://doi.org/10.1016/j.tics.2012.08.004>
- Rakoczy, H., & Oktay-Gür, N. (2020). Why Do Young Children Look so Smart and Older Children Look so Dumb on True Belief Control Tasks? An Investigation of Pragmatic Performance Factors. *Journal of Cognition and Development*, 21(2), 213–239. <https://doi.org/10.1080/15248372.2019.1709467>
- Rubio-Fernández, P., & Geurts, B. (2013). How to pass the false-belief task before your fourth birthday. *Psychological Science*, 24(1), 27–33. <https://doi.org/10.1177/0956797612447819>
- Ruffman, T., Garnham, W., Import, A., & Connolly, D. (2001). Does eye gaze indicate implicit knowledge of false belief? Charting transitions in knowledge. *Journal of Experimental Child Psychology*, 80(3), 201–224. <https://doi.org/10.1006/jecp.2001.2633>
- Schidelko, L. P., Proft, M., & Rakoczy, H. (2022). How do children overcome their pragmatic performance problems in the true belief task? The role of advanced pragmatics and higher-order theory of mind. *PLoS ONE*, 17(4), e0266959. <https://doi.org/10.1371/journal.pone.0266959>
- Southgate, V., Senju, A., & Csibra, G. (2007). Action anticipation through attribution of false belief by 2-year-olds. *Psychological Science*, 18(7), 587–592. <https://doi.org/10.1111/j.1467-9280.2007.01944.x>
- Wenzel, L., Dörrenberg, S., Proft, M., Liszkowski, U., & Rakoczy, H. (2020). Actions do not speak louder than words in an interactive false belief task. *Royal Society Open Science*, 7(10). <https://doi.org/10.1098/rsos.191998>
- Zhang, Y., Hedo, R., Rivera, A., Rull, R., Richardson, S., & Tu, X. M. (2019). Post hoc power analysis: Is it an informative and meaningful analysis? *General Psychiatry*, 32(4), e100069. <https://doi.org/10.1136/gpsych-2019-100069>

Appendix C

Dear Dr. de Klerk,

Thank you very much for the decision letter regarding our manuscript. We are very happy to hear that the manuscript has been accepted for publication, although minor modifications are still required. We are very grateful for the opportunity to address these remaining issues. Please find attached the revised version. Below, we detail how we responded to each issue (in red).

Looking forward to hearing back from you.

Kind regards,

Pamela Barone

Associate Editor Comments to Author (Dr Carina de Klerk):

Dear Dr. Barone,

Thank you for resubmitting your revised manuscript. I just have a few minor remaining comments that would need to be addressed before the manuscript can be published.

In some places in the manuscript the research questions are 'too big' relative to what the study does. The research questions relate to the state of the field while the manuscript focuses on one study. It is true that the task used in his manuscript is particularly informative, but the current study's results are not necessarily going to generalise to all other tasks in the field. I do think it's great to raise these questions in the introduction section as the theoretical rationale for the study, but the way they are presented at the beginning of the discussion section and summary paragraph may give the wrong impression that this single study is going to be able to provide a conclusive answer.

Response: Thank you for pushing us in making clearer the scope of these questions. We now toned down the way our study relates to these questions, both in the abstract and in the discussion.

Please clarify in the main manuscript how the ignorance condition was introduced. I.e. how did you make it clear to the child that the bear didn't witness the original placement and transfer. It seems important to describe this in the main manuscript because the interaction between the experimenter and bear is described as a possible explanation for the findings in the discussion section (e.g. on page 19 it is stated that "Another potential concern regarding the present study is that the IG

condition may have been confusing (...). On the other hand, the way E1 and the bear interact may be taken to suggest that E1, something like an omniscient narrator in the scene, expects the bear to know where the object is”).

Response: Thank you for this suggestion. In the revised version of the main manuscript, we explained the ignorance condition in more detail.

Relatedly, in their reply to the reviewer’s comments on the previous version of their manuscript the authors mentioned that you discuss how future studies may implement more refined versions of the ignorance conditions. It would be good if the authors could do this in a more explicit manner, i.e. acknowledge why exactly the IG condition may have been confusing and what could be done to remedy this in the future.

Response: Thank you for insisting on this critical issue. In response, we now try to make these points more explicit.

In the discussion section the authors suggest that the pattern of results in the ignorance condition may have resulted from some form of ‘pull of the real’ and that in cases of uncertainty it may be safer to go for the objectively good than for the objectively bad answer. Could a reality bias in some of the children also have influenced performance in the FB condition?

Response: Thank you for coming back to this issue. In fact, it could influence the performance of younger children in the FB condition. Our results show that young children’s looking behavior (first look and DLS) in the FB condition is not different from chance level, while older children looked at the empty box in both looking measures. The reality bias could partially explain this age group difference but only for looking behavior. Children’s choice of boxes though their interactive behavior was not different from chance in either group.

Like reviewer 2 requested please indicate how many of the effects were significant rather than describing results as ‘largely non-significant’ or ‘largely showing no differences’.

Response: Thank you for stating this issue again.

We are not quite sure, though, how to read this point. If the suggestion refers to the comparison between FB and IG conditions and which of the effects are significant,

we have now modified the second paragraph of the discussion to address how many of these effects were significant (1 out of 3).

However, if this suggestion refers to the three questions more generally we would like to address in our manuscript, then the picture is the following.

Q1: How robust and reliable are anticipatory looking and interactive measures of early FB understanding? = We found a significant effect (difference between TB and FB in anticipatory looking and interactive behavior). We did not find an effect regarding uncertainty measures.

Q2: Do these measures actually tap FB understanding? = Here, the FB-IG differences mentioned above are relevant.

Q3: Do they really tap an earlier and more basic form of FB understanding than explicit standard tasks? = Here, the differences between implicit and explicit measures are relevant, in particular regarding the question whether those children who fail to pass the standard FB tasks nevertheless show competence in the anticipatory measures (i.e. perform differently in FB from TB and also from IG).

If it refers to the three questions, then we indicate at the relevant places in the manuscript which effects were and which were not significant.

Please avoid using abbreviations unless they are very commonly used (like FB or TOM). So please just refer to anticipatory looking instead of AL (if possible within the word count).

Response: Thanks. We have removed the abbreviation AL and used anticipatory looking instead.

Note that the acknowledgements section refers to 'assistants' to CogSci and BCCCD which presumably is meant to be 'attendants'.

Response: Thanks for identifying this mistake.

Appendix D

Dear Dr. de Klerk,

Thank you very much for the decision regarding our manuscript. We are very happy to hear that the manuscript has been accepted for publication, although minor modifications are still required. Please find attached the revised version.

Looking forward to hearing back from you.

Kind regards,

Pamela Barone

Associate Editor Comments to Author (Dr Carina de Klerk):

Associate Editor Comments to Author (Dr Carina de Klerk):

Associate Editor

Comments to the Author:

Please edit the first sentence of the abstract. I understand that this change was made in response to my comment about the scope of the research questions addressed by this work, but I am not sure it makes sense to suggest that the study aims to contribute to clarifying three questions. The questions aren't being clarified by the work right? Instead the work aims to bring the field one step closer to answering these questions. Possible alternatives: "This paper aimed to contribute to answering three questions:" or "This paper was inspired by three questions currently occupying the field of infant theory of mind understanding". Please also update the final section of the introduction section based on my previous comment relating to the scope of the paper/research questions. The current work is still presented as if it can provide a conclusive answer to these big questions. While it is an informative study it is not going to 'solve' or answer these questions conclusively.

Response: Thank you for your suggestions regarding the way we should frame this important issue in our paper.

In the abstract, we used the first alternative you offered due to the word limit. The final section of the introduction was modified accordingly.